# FAITHFUL EXPLANATIONS OF BLACK-BOX NLP MODELS USING LLM-GENERATED COUNTERFACTUALS

**Yair Ori Gat**[T*]**, Nitay Calderon**[T*]**, Amir Feder**[C]**,**
**Alexander Chapanin**[T]**, Amit Sharma**[M] **and Roi Reichart**[T]
[T]Faculty of Data and Decision Sciences, Technion, IIT
[C]Columbia University Data Science Institute, [M]Microsoft Research India

## ABSTRACT

Causal explanations of the predictions of NLP systems are essential to ensure safety and establish trust. Yet, existing methods often fall short of explaining model predictions effectively or efficiently and are often model-specific. In this paper, we address model-agnostic explanations, proposing two approaches for counterfactual (CF) approximation. The first approach is CF generation, where a large language model (LLM) is prompted to change a specific text concept while keeping confounding concepts unchanged. While this approach is demonstrated to be very effective, applying LLM at inference-time is costly. We hence present a second approach based on matching, and propose a method that is guided by an LLM at training-time and learns a dedicated embedding space. This space is faithful to a given causal graph and effectively serves to identify matches that approximate CFs. After showing theoretically that approximating CFs is required in order to construct faithful explanations, we benchmark our approaches and explain several models, including LLMs with billions of parameters. Our empirical results demonstrate the excellent performance of CF generation models as model-agnostic explainers. Moreover, our matching approach, which requires far less test-time resources, also provides effective explanations, surpassing many baselines. We also find that Top-K techniques universally improve every tested method. Finally, we showcase the potential of LLMs in constructing new benchmarks for model explanation and subsequently validate our conclusions. Our work illuminates new pathways for efficient and accurate approaches to interpreting NLP systems.

## 1 INTRODUCTION

Providing faithful explanations for Natural Language Processing (NLP) model predictions is imperative to ensure safe deployment, establish trust, and foster scientific discoveries (Amodei et al., 2016; Goodman & Flaxman, 2017; Guidotti et al., 2019; Jacovi & Goldberg, 2020; Lyu et al., 2022). These aspects are particularly significant in NLP, where the complexity of language and the opaque behavior of black-box models. For an explanation to be genuinely faithful and accurately depict a model's underlying reasoning, it is crucial to establish causality. Recognizing this inherent link and following previous works (Vig et al., 2020; Geiger et al., 2020; Feder et al., 2021b; Wu et al., 2023a; Feder et al., 2023; Wu et al., 2023b), in this paper we introduce a theoretical framework that binds faithfulness and causal explanation together and propose two causal-inspired explanation methods.

In contrast to model explanation techniques that *often conflate correlation with causation*, causal-inspired methods usually contrast predictions for an input example with those of its counterfactual (Soulos et al., 2020; Elazar et al., 2021; Finlayson et al., 2021). Indeed, counterfactuals are at the highest level of Pearl's causal hierarchy (Pearl, 2009), highlighting how changes lead to a different prediction. However, they cannot be acquired without knowing the complete data-generating process (or structural model) of the text (Balke & Pearl, 1994), which is not practical for a given real-world problem. Hence, we turn to *approximated counterfactuals (CFs)*: Imagining how a given text would look like if a certain concept in its generative process were different (Calderon et al., 2022).

---

*Equal contribution. Corresponding author: nitay@campus.technion.ac.il
Code: https://github.com/YairGat/causal-explanations

In §3.1, we propose a simple, intuitive, and essential criterion for explanation methods: *Order-faithfulness* – If the explanation method ranks one concept's impact above another's, its true causal effect should genuinely be greater. We then theoretically show that in contrast to non-causal explanation methods (that overlook confounders, for example), approximating CF methods are always order-faithful. Moreover, since CFs do not depend on the explained model, they enable causal estimation in a *model-agnostic manner*. Model-agnostic explanation methods offer numerous advantages, especially during model selection, debugging, and deployment. For example, developers juggling multiple models can effortlessly rank them based on their vulnerability to confounding biases (such as gender bias). Hence, we focus only on model-agnostic explanations in this study.

Yet, acquiring CFs in past work was limited to simple local manipulations (Ribeiro et al., 2020; Ross et al., 2021; Wu et al., 2021) or costly manual annotation (Gardner et al., 2020; Kaushik et al., 2020a), hindering practical causal effect estimation of high-level concepts on NLP models. To address the above limitations, we rely on Large Language Models (LLMs) and introduce two practical approaches for model explanation. The first approach is *Counterfactual Generation*, in which an LLM is prompted to change a concept of a text while holding confounding concepts fixed. We empirically demonstrate that *LLM-generated CFs are the SOTA explanations*. However, utilizing LLMs comes with a high computational and financial cost. Moreover, the slow generation process may limit the application of CF generation methods, especially in scenarios demanding real-time explanations or when explaining vast quantities of data (Calderon et al., 2023a).

Therefore, we introduce an efficient alternative: *Matching*, which is up to 1000 times faster (see Appendix §B). We use matching to estimate the causal effect, pairing each treated unit with one or more control units with similar observed characteristics, i.e., identifying CF approximations within a pre-defined candidate set. To enhance matching quality, we propose in §3.3 a novel method for learning a causal embedding space. Given a causal graph (e.g., Figure 1), we first employ an LLM for generating CFs (note that the LLM is used only during the learning phase). We also identify possible valid matches: Instances with identical values in observable adjustment variables (that satisfy the back-door criterion). Subsequently, we train our causal representation model by minimizing an objective composed of contrastive loss components. The causal representation model learns to encode examples similarly to their LLM-generated CFs or matched examples. Conversely, it is trained to produce dissimilar representations for the query example and any misspecified CFs. As a result, representations remain faithful to the causal graph and can effectively identify matching candidates.

We undertake rigorous experiments to compare our causal language representation method against various generative and matching baselines. For this purpose, we employ CEBaB (Abraham et al., 2022), the only established benchmark for evaluating causal explanation methods. We estimate the causal effect of 24 concept interventions on the predictions of five models, three small fine-tuned LMs, and two zero-shot LLMs with billions of parameters. In Appendix §C we conduct an ablation study to examine the impact of each design choice within our matching method.

Our empirical results reveal three key findings: (1) The counterfactual generation approach provides strong black-box model explanations. (2) Our method for learning causal representations for matching outperforms all the matching baselines, including the best-performing method from the CEBaB paper (see Figure 6). (3) Top-$K$ matching universally greatly enhances the explanatory capabilities of all examined methods, including generative models, when generating multiple CFs.

Finally, building on our findings that LLMs can produce high-quality CFs, we address a pressing challenge: The lack of benchmarks for NLP model explanation methods. In §D we demonstrate how to construct such a benchmark by leveraging GPT-4 to generate pairs of inputs and their CFs (used as "golden CF"). The new dataset focuses on Stance Detection, an NLP task that determines the position expressed by the speaker towards a particular topic (e.g., abortions, climate change) (Küçük & Can, 2021). By utilizing it, we observe similar patterns to our main conclusions. We hope this study will pave the way for safer, more transparent, and more accountable AI systems.

## 2 RELATED WORK

**Explanation methods and causality.** Feature importance tools (Molnar, 2020) and gradient-based methods methods (Zeiler & Fergus, 2014; Binder et al., 2016; Shrikumar et al., 2017; Gat et al., 2022) often restricted to input features, making it challenging to connect their analyses with real-world

concepts that do not reduce to simple properties of inputs. In this study, we focus on the explanation of high-level real-world concepts (Poeta et al., 2023). In probing, a small supervised (Conneau et al., 2018; Tenney et al., 2019) or unsupervised (Clark et al., 2019; Manning et al., 2020; Saphra & Lopez, 2019) model is used to measure whether specific concepts are encoded at specific places in a network. Geiger et al. (2021) show that probes cannot reliably provide causal explanations. Other methods perturb input or hidden representations to create CF states that can then be used to estimate causal effects (Ribeiro et al., 2016; Elazar et al., 2021; Finlayson et al., 2021; Soulos et al., 2020; Vig et al., 2020; Geiger et al., 2021). However, these methods are prone to generating implausible inputs or network states unless the interventions are carefully controlled (Geiger et al., 2020). Recently, Wu et al. (2023a) proposed the Causal Proxy Model, a novel explanation method inspired by S-learner (Künzel et al., 2019), which mimics the behavior of the explained model when the input is a CF. Although CPM is effective, the need to train a distinct explainer for each explained model is a major disadvantage. Conversely, our study focuses on model-agnostic explanation methods.

**Approximating counterfactuals.** A common use-case for CF examples in machine learning is for data augmentation (Hong et al., 2023). These CFs involve perturbations to confounding factors (Garg et al., 2019), or to the label (Kaushik et al., 2019; 2020b; Jha et al., 2020). CF examples can be generated through manual editing, heuristic keyword replacement, or automated text rewriting (Kaushik et al., 2019; Gardner et al., 2020; Shekhar et al., 2017; Garg et al., 2019; Feder et al., 2021a; Zmigrod et al., 2019; Riley et al., 2020; Wu et al., 2021; Mao et al., 2021; Rosenberg et al., 2021). Manual editing is accurate but expensive, while keyword-based methods can be limited in coverage and difficult to generalize across languages (Antoniak & Mimno, 2021; Zhou & Wu, 2023). While LLMs can overcome these challenges, they come with prolonged latency, high financial costs, or privacy constraints. Our matching approach is an efficient alternative to overcome these limitations.

## 3 METHOD

This study focuses on black-box NLP model explanations by estimating the causal effect of high-level concepts on model prediction. The first requirement for any causal estimation method is access to a *causal graph* that describes our causal beliefs, i.e., the concepts and the relationships between them (e.g., the causal graph in Figure 1). See Appendix §F.1 for further discussion about the requirements.

To provide an accurate explanation, we should estimate the *Average Treatment Effect (ATE)* (Pearl, 2009) or the *Individual Treatment Effect (ITE)* (Shpitser & Pearl, 2006; Shalit et al., 2017). Specifically, the treatment is a high-level concept influencing the text (such as *ambiance* in a restaurant review), and the outcome variable is a prediction of a text classifier. When discussing model explanation in the sense of the causal effect of a concept on the model prediction, the common versions of the ATE and the ITE are the *Causal Concept Effect (CaCE)* (Goyal et al., 2019) and the *Individual Causal Concept Effect (ICaCE)* (Abraham et al., 2022), which we next formally define.

Given a DGP $\mathcal{G}$, an intervention on a treatment variable $T : t \rightarrow t'$ (for simplicity we sometimes write $T \leftarrow t'$), a model $f$, and a query example $\boldsymbol{x}_t$, the CaCE and ICaCE are defined to be:

$$\mathrm{CaCE}_f(T, t, t') = \mathbb{E}_{\boldsymbol{x}' \sim \mathcal{G}} \left[ f(\boldsymbol{x}') | do(T = t') \right] - \mathbb{E}_{\boldsymbol{x}' \sim \mathcal{G}} \left[ f(\boldsymbol{x}') | do(T = t) \right] \tag{1}$$

$$\mathrm{ICaCE}_f(\boldsymbol{x}_t, T, t') = \mathbb{E}_{\boldsymbol{x}' \sim \mathcal{G}} \left[ f(\boldsymbol{x}') | \boldsymbol{x}_t, do(T = t') \right] - f(\boldsymbol{x}_t)$$

One way to estimate the causal effect and explain the model is by using counterfactuals, which enables causal estimation in a model-agnostic manner (because they do not depend on the explained model). While we are aware of the causal graph that encodes our causal beliefs, for a given problem, we cannot access the complete *data-generating process (DGP)* nor the variable values of an example, and therefore, we cannot produce gold counterfactuals. Given this, we propose two approaches for approximating them: Counterfactual generation (§3.2) and Matching (§3.3).

If $\widetilde{\boldsymbol{x}}_{t'}$ is an *approximated counterfactual (CF)* of the text $\boldsymbol{x}_t$ resulting from the intervention $T : t \rightarrow t'$ (can be either a model-generated CF or a match), then the causal effect estimators are:

$$\widehat{\mathrm{ICaCE}}_f(\boldsymbol{x}_t, T, t, t') := \widehat{\mathrm{ICaCE}}_f(\boldsymbol{x}_t, \widetilde{\boldsymbol{x}}_{t'}) = f(\widetilde{\boldsymbol{x}}_{t'}) - f(\boldsymbol{x}_t) \tag{2}$$

$$\widehat{\mathrm{CaCE}}_f(T, t, t') = \frac{1}{|\mathbb{D}|} \left( \sum_{\boldsymbol{x}_t} \widehat{\mathrm{ICaCE}}_f(\boldsymbol{x}_t, \widetilde{\boldsymbol{x}}_{t'}) + \sum_{\boldsymbol{x}_{t'}} \widehat{\mathrm{ICaCE}}_f(\widetilde{\boldsymbol{x}}_t, \boldsymbol{x}_{t'}) + \sum_{\boldsymbol{x}} \widehat{\mathrm{ICaCE}}_f(\widetilde{\boldsymbol{x}}_t, \widetilde{\boldsymbol{x}}_{t'}) \right)$$

### 3.1 COUNTERFACTUALS AS AN IDEAL MODEL EXPLANATION

While definitions of faithfulness vary across literature, there is a consensus that a faithful explanation method should *accurately describe a model's underlying reasoning* (Rudin, 2019; Dasgupta et al., 2022; Lyu et al., 2022). Relying on the causality literature, we propose a simple, intuitive, and essential criterion: *Order-faithfulness*. This implies that if the explanation method ranks the impact of one concept (or intervention) higher than another, the true causal effect of the first concept should exceed that of the second. Order-faithfulness is crucial in scenarios like model selection and deployment. For instance, to determine the fairness of a hiring model, it is essential to know whether it incorrectly prioritizes non-relevant concepts (e.g., gender) over genuine concepts (e.g., experience).

**Definition** (Order-Faithfulness). *An explanation method $S$ is **order-faithful** if:*

$$\mathbb{E}_{\mathbb{D}\sim\mathcal{G}}[S(f, T_1, t_1, t_1')] > \mathbb{E}_{\mathbb{D}\sim\mathcal{G}}[S(f, T_2, t_2, t_2')] \iff \mathsf{CaCE}_f(T_1, t_1, t_1') > \mathsf{CaCE}_f(T_2, t_2, t_2') \quad (3)$$

Note that the definition above does not mandate the explanation method to estimate the causal effect accurately or generate scores proportional to the causal effects. Instead, it simply requires the method to be rank-preserving. In addition, it holds for comparing the importance of two distinct concepts, as many attribution methods like LIME do (Ribeiro et al., 2016), and it also holds when assessing the importance of changes in the same concept. We believe that order-faithfulness is a necessary condition for the broad definition of faithfulness. If an explanation method ranks the importance of one concept above another, but in fact its causal effect is smaller, then the method does not accurately reflect the actual reasoning process of the model being explained.

In Appendix §A we provide detailed formal definitions of two explanation methods: Approximated CF explanation methods and Non-causal explanation methods (which we define as any function of the observed data). Furthermore, we prove the following theorem, which elucidates why approximated CF methods are always order-faithful, unlike non-causal ones.

**Theorem.** *For an explained model $f$, (1) The approximated CF explanation method $S_{CF}$ is order-faithful for every DGP $\mathcal{G}$ and a pair of interventions; (2) For every DGP $\mathcal{G}$, there exist a DGP $\mathcal{G}'$ resulting from a small modification of $\mathcal{G}$, a model $f'$, and a pair of interventions $T_1 : t_1 \to t_1', T_2 : t_2 \to t_2'$ such that the non-causal explanation method $S_{NC}$ is not order-faithful.*

*Proof sketch*: We first prove that approximated CF methods are always faithful by showing the expected prediction of an approximated CF is equal to the interventional one (conditioned on the do operator). To prove the second part, we construct $\mathcal{G}'$ by introducing a new unobserved confounder variable (the "small modification") that reverses the order of two interventions' causal effect on a new model $f'$. In addition, we ensure that the non-causal method produces the same explanations as it produces for $\mathcal{G}$ since the generation process of $\boldsymbol{x}$ is the same in the two DGPs. Hence, $S_{NC}$ cannot be order-faithful in $\mathcal{G}$ and $\mathcal{G}'$ at the same time.

The theorem underscores the importance of developing causal-inspired explanation methods and suggests why our CF-based approaches are more effective than other baselines.

### 3.2 LLM-GENERATED COUNTERFACTUALS

Our first approach for CF approximation is *Counterfactual Generation*, in which a generative model (an LLM) is prompted with to change an attribute conveyed in the text. We explicitly inject our causal beliefs into the prompt using the fundamental causal concept of *adjustment* (see Appendix §H).

Given a query example $\boldsymbol{x}_t$, we instruct the LLM to intervene in the text by changing the value of the concept $T$ from $t$ to $t'$. Additionally, we instruct the LLM to consider the adjusted concepts (confounders) and hold them fixed. Optionally, we can specify the non-adjusted concepts (such as mediators and colliders) if they exist and ask the LLM to consider that the intervention on $T$ might affect these concepts. This specification can increase the precision of the CF. For the Top-K technique, we simply generate multiple CFs. See Appendix §F.2 for examples of prompts and generated CFs.

Nevertheless, utilizing an LLM at inference time is frequently impractical because of its prolonged latency, high financial costs, or privacy concerns that prevent data from being sent to external servers (see Calderon et al. (2023a) and our latency measurements in Appendix §B). Considering these challenges, in the next subsection we introduce an efficient alternative, the *matching* technique.

Table 1: Toy illustration of examples (and their concept values) sampled from the four sets of a given query example and an intervention of changing service to positive ($S \leftarrow +$). The misspecified CF ($\mathbb{X}_{\neg\mathsf{CF}}$) resulted from a wrong intervention of removing the ambiance concept ($A \leftarrow ?$).

| Query | Excellent lobster and decor, but rude waiter. | | | |
|---|---|---|---|---|
| | $F : +$ | $S : -$ | $N : ?$ | $A : +$ |
| $\mathbb{X}_{\mathsf{CF}}$ | Excellent lobster and decor, and friendly waiter. | | | |
| | $F : +$ | $S : +$ | $N : ?$ | $A : +$ |
| $\mathbb{X}_{\mathsf{M}}$ | A cozy atmosphere with great pizza and service. | | | |
| | $F : +$ | $S : +$ | $N : ?$ | $A : +$ |
| $\mathbb{X}_{\neg\mathsf{CF}}$ | Excellent lobster and rude waiter. | | | |
| | $F : +$ | $S : +$ | $N : ?$ | $A : ?$ |
| $\mathbb{X}_{\neg\mathsf{M}}$ | Too expensive for a loud chaos with bland food. | | | |
| | $F : -$ | $S : ?$ | $N : -$ | $A : -$ |

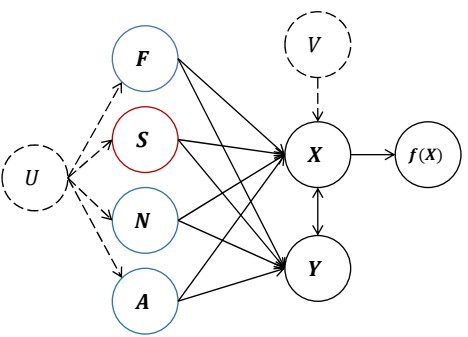

Figure 1: Our illustration of the causal graph of the CEBaB benchmark (Abraham et al., 2022). $U$ and $V$ are (unobserved) exogenous variables representing the state of the world. For example, $V$ may account for style, syntax, and length. The four concepts $F$ (food), $S$ (service), $A$ (ambiance) and $N$ (noise) affect both the textual review $X$ and its five-star rating $Y$. The relationship between $X$ and $Y$ can be causal or anti-causal. $f$ is the model we wish to explain, which is trained to predict $Y$ from $X$. In this study, we are interested in estimating the causal effect of changing a concept on the predictive model $f$ (e.g., changing the value of the red variable $S$ from positive to negative).

## 3.3 CAUSAL REPRESENTATION LEARNING FOR MATCHING

Although counterfactual generation is a valuable model explanation approach, employing LLMs during inference can be infeasible. Conversely, there is an opportunity to harness model-generated CFs to train an efficient method that seeks approximations within a pre-defined set of candidate texts, mitigating the need for direct generation. This approach is known as *matching* (Stuart, 2010) and endeavors to match each treatment unit with its most similar control unit. To allow effective matching, we introduce a novel method that learns to encode texts in a space faithful to the causal graph, i.e., each treatment unit resonates closely with its CF counterparts and remains distinct from confounding texts. Formally, a matched example of $\boldsymbol{x}_t$ is defined to be:

$$m_1(\boldsymbol{x}_t) = \underset{\boldsymbol{x}_{t'} \in \mathbb{D}(T=t')}{\arg\max} \; s\left(\phi(\boldsymbol{x}_t), \phi(\boldsymbol{x}_{t'})\right) \tag{4}$$

The *candidate set* is denoted by $\mathbb{D}(T = t')$ and is also known as the control group, contains only texts whose treatment variable equals $t'$, and is known in advance (i.e., both the treatment and control groups are pre-defined and there is no need to infer which text belong to which group). $\phi(\cdot)$ is a feature extractor function (the language representation model we aim to learn) that maps discrete text to the hypersphere space where matching is conducted by ranking candidates according their similarity with the query. In our experiments, the similarity function $s(\cdot)$ is the cosine similarity (defined in Eq. 11). Notice that Eq 4 defines *1:1 matching*, for *Top-K matching* we use: $\frac{1}{K}\sum_{i=1}^{K} \widehat{\mathrm{ICaCE}}_f(\boldsymbol{x}_t, m_i(\boldsymbol{x}_t))$.

Our causal representation method is based on an Encoder-only language model (e.g., RoBERTa (Liu et al., 2019) or a SentenceTransformer (Reimers & Gurevych, 2019)) fine-tuned to minimize an objective consisting of six contrastive loss components. The goal of the objective is to enhance the similarity with approximate CF candidates while reducing similarity with misspecified CFs.

Given a causal graph, an intervention on a treatment variable $T : t \rightarrow t'$ and an example $\boldsymbol{x}_t$, we define the four following sets (see toy examples in Table 1):

1. $\mathbb{X}_{\mathsf{CF}}$ - This set contains *Counterfactuals*, obtained by modifying $\boldsymbol{x}_t$ through the intervention on $T$, while the other variables are held unchanged. We utilize an LLM (ChatGPT) to simulate several CFs (up to ten).

2. $\mathbb{X}_{\mathsf{M}}$ - This set contains *Matched samples*. These instances share identical values for the adjustment variables (that satisfy the back-door criterion, see Appendix §H). Our study employs a smaller model to predict their values and construct a set of valid matches.

3. $\mathbb{X}_{\neg\mathsf{CF}}$ - This set contains *Misspecified Counterfactuals*. These examples are derived from editing $\boldsymbol{x}_t$ through interventions on variables other than the treatment variable. The term *misspecification* is well-known in the causal literature and indicates a wrong assumption of the DGP (Vansteelandt et al., 2012). In our case, a wrong intervention.

4. $\mathbb{X}_{\neg\mathsf{M}}$ - This set contains *Misspecified Matched samples*. This is the complementary set to $\mathbb{X}_{\mathsf{M}}$, containing all those instances that do not qualify as valid matches due to differing values of the adjustment variables.

The causal representation model is trained by minimizing the sum of six contrastive loss components:

$$
\begin{aligned}
\mathcal{L}(\boldsymbol{x}_t) = \; & \mathcal{L}(\boldsymbol{x}_t, \mathbb{X}_{\mathsf{CF}}, \mathbb{X}_{\neg\mathsf{M}}) + \mathcal{L}(\boldsymbol{x}_t, \mathbb{X}_{\mathsf{CF}}, \mathbb{X}_{\neg\mathsf{CF}}) + \mathcal{L}(\boldsymbol{x}_t, \mathbb{X}_{\mathsf{CF}}, \mathbb{X}_{\mathsf{M}}) \\
& + \mathcal{L}(\boldsymbol{x}_t, \mathbb{X}_{\mathsf{M}}, \mathbb{X}_{\neg\mathsf{M}}) + \mathcal{L}(\boldsymbol{x}_t, \mathbb{X}_{\mathsf{M}}, \mathbb{X}_{\neg\mathsf{CF}}) \\
& + \mathcal{L}(\boldsymbol{x}_t, \mathbb{X}_{\neg\mathsf{CF}}, \mathbb{X}_{\neg\mathsf{M}})
\end{aligned}
\tag{5}
$$

Where the *contrastive loss* (Wang & Liu, 2021) is defined below and aims to attract some input $\boldsymbol{x}$ to the "positive set" $\mathbb{X}_+$, and separate it from the "negative set" $\mathbb{X}_-$ ($\tau$ is a temperature hyperparameter):

$$
\mathcal{L}(\boldsymbol{x}, \mathbb{X}_+, \mathbb{X}_-) = -\log\left[\frac{\sum_{\boldsymbol{x}_+}\exp(s(\phi(\boldsymbol{x}), \phi(\boldsymbol{x}_+))/\tau)}{\sum_{\boldsymbol{x}_+}\exp(s(\phi(\boldsymbol{x}), \phi(\boldsymbol{x}_+))/\tau) + \sum_{\boldsymbol{x}_-}\exp(s(\phi(\boldsymbol{x}), \phi(\boldsymbol{x}_-))/\tau)}\right]
\tag{6}
$$

We hypothesize (and verify) that omitting components from the objective (Eq. 5) risks favoring inaccurate matches, as we highlight in the ablation study at Appendix §C. Our objective prioritizes text matches that closely resemble CF approximations within a candidate set, defaulting to adjusting variable values in their absence. When choosing between misspecified CFs or misspecified matches, the methods favor the latter, given their potential shared traits with the CF, such as syntax or style linked to exogenous variables. Formally, given a query example $\boldsymbol{x}_t$ and four matching candidates $\boldsymbol{x}_{\mathsf{CF}} \in \mathbb{X}_{\mathsf{CF}}, \boldsymbol{x}_{\mathsf{M}} \in \mathbb{X}_{\mathsf{M}}, \boldsymbol{x}_{\neg\mathsf{CF}} \in \mathbb{X}_{\neg\mathsf{CF}}$, the candidates' ranking, which is based on their similarity with $\boldsymbol{x}_t$, should be $\boldsymbol{x}_{\neg\mathsf{M}} \preceq \boldsymbol{x}_{\neg\mathsf{CF}} \preceq \boldsymbol{x}_{\mathsf{M}} \preceq \boldsymbol{x}_{\mathsf{CF}}$. This desirable order is not arbitrary and is a direct product of the six components, ensuring robustness to variations of the candidate set. In addition, we investigate in the ablation study (Appendix §C) the applicability of our matching method without pre-identified concept annotations in the training dataset (i.e., completely unsupervised setup). To this end, we utilize an LLM and predict the concept values in a zero-shot manner.

In Appendix §G we provide additional implementation details, including the training procedure of our causal representation model (§G.2) and hyperparameters (§G.3).

## 4 EXPERIMENTAL SETUP

### 4.1 MODEL EXPLANATION PIPELINE

**Causal Estimation-Based Benchmark (CEBaB).** CEBaB (Abraham et al., 2022) is an interventional dataset consisting of examples, interventions, and a corresponding human-written ground-truth CF for each example and intervention. CEBaB originated from thousands of original restaurant reviews obtained from OpenTable. Each entry (original review or CF) in CEBaB received a 5-star sentiment rating, labeled by five annotators. Furthermore, every text was annotated at a conceptual level, as positive, negative or neutral w.r.t. four central mediating concepts: Food, Service, Ambiance, and Noise (see Figure 1). CEBaB contains two train sets, exclusive ($N = 1463$) and inclusive (which we do not use), development ($N = 1672$), and test ($N = 1688$) sets. We randomly split the exclusive set into two equal sets: the train set and a matching candidates set.

**Explanation method evaluation.** For a given example $\boldsymbol{x}_t$ and an intervention $T : t \to t'$, we can estimate the golden $\mathtt{ICaCE}$ using the corresponding human-written CF: $\widehat{\mathtt{ICaCE}}_f(\boldsymbol{x}_t, \widetilde{\boldsymbol{x}}_{t'}^h)$. This is the difference between the outputs of the model $f$ (a five-dimension vector in CEBaB corresponding to the sentiment). Likewise, in a CF-based explanation method, the estimation is computed with the approximated CF: $\widehat{\mathtt{ICaCE}}_f(\boldsymbol{x}_t, \widetilde{\boldsymbol{x}}_{t'}^m)$. We can then evaluate the explanation method using:

$$
\mathtt{Err}(f, m, T, t, t') = \frac{1}{|\mathbb{D}(T \leftarrow t')|}\sum_{(\boldsymbol{x}_t, \widetilde{\boldsymbol{x}}_{t'}^h)\in\mathbb{D}}\mathtt{Dist}\left(\widehat{\mathtt{ICaCE}}_f(\boldsymbol{x}_t, \widetilde{\boldsymbol{x}}_{t'}^h), \widehat{\mathtt{ICaCE}}_f(\boldsymbol{x}_t, \widetilde{\boldsymbol{x}}_{t'}^m)\right)
\tag{7}
$$

Where Dist can be, for example, the $L2$ distance, the cosine distance $(1 - \cos)$, or the norm difference (see Abraham et al. (2022) and Eq. 12). The Err scores we present in this study are the mean over 24 interventions: four concepts with three values (six interventions for each concept). See Appendix §G.1 and §G.4 for full details about the causal estimation and explanation evaluation pipelines.

## 4.2 MODELS AND BASELINES

In Appendix G we provide additional implementation details, including hyparameters and prompts.

**Generative approach.** We employ three different CF generation techniques: *Zero-shot Generative*, *Few-shot Generative* (with three demonstrations) and *Fine-tune Generative*. The first two are based on the Decoder-only ChatGPT (Ouyang et al., 2022), while the third is based on an Encoder-decoder T5-base model (Raffel et al., 2019) trained using a parallel dataset, comprised of original reviews and their human-written CFs (the inclusive train set of CEBaB). The zero-shot and few-shot prompts are described in Appendix §I.

**Matching baselines.** We benchmark against six matching methods: (1) *Random Match*, which randomly selects an example from the candidate set; (2) *Propensity* (Benedetto et al., 2018), which first estimates the propensity score $P(T = t'|x)$ using a concept predictor, and then conducts matching based on this score; (3) *Approx*, which randomly selects an example from $\mathbb{X}_M$. Notice that $\mathbb{X}_M$ can be an empty set, thus, unlike the other baselines, matching may not be performed; (4) *PT RoBERTa*, which utilizes a pre-trained LM to represent texts and finds matches using their cos similarity; (5) *PT S-Transformer*, same as PT RoBERTa, expect the backbone model is a SentenceTransformer model (Reimers & Gurevych, 2019) trained to maximize semantic textual similarity; and (6) *FT S-Transformer*, a SentenceTransformer fine-tuned to predict the five-star sentiment;
Notice that Abraham et al. (2022) found that Approx outperforms all seven tested (non-matching) baselines (see Figure 6). Therefore we do not include them in this study.

**Interpreted models.** We interpret five different models of varying sizes. The first three are Encoder-only models fine-tuned using only the original reviews from the train set of CEBaB to predict the five-star sentiment. These models include DistilBERT (Sanh et al., 2019), BERT (Devlin et al., 2019), and RoBERTa (Liu et al., 2019). The other two are the zero-shot chat versions of the Decoder-only Llama-2 model (Touvron et al., 2023) with 7 and 13 billion parameters. We extract the five-star sentiment distribution of the Llama models using the prompt described in Figure 7.

## 5 RESULTS

Our main results are provided in Table 2. In Appendix §C we provide a thorough ablation study, demonstrating the effect of each of our design choices, including, among others, the impact of variations of the candidate set and the causal model objective (Eq. 5). Our three main findings are:

**1. Counterfactual generation models are the SOTA model-agnostic explainers.** The first promise of this study is that CF generation models are strong explainers. As can be seen from Table 2, the generative models achieve much lower errors compared to the other methods, making them the SOTA model-agnostic explainers. In addition, the table provides two additional findings: (1) Few-shot prompts are better than zero-shot prompts for generating CFs; and (2) The small fine-tuned Encoder-decoder T5 is the best generative model. This suggests that when a parallel dataset consisting of pairs of input examples and their human-written CFs is available, then it is better to fine-tune a small model. Moreover, the small model is computationally (and financially) less expensive than LLMs (but much slower than matching methods). Nevertheless, consider that collecting a parallel dataset for fine-tuning is labor intensive.

**2. The causal representation model is the best-performing matching method.** Table 2 also sheds light on the performance of our novel causal representation method (described in §3.3, *our causal model* in short) in comparison to various explainers and matching baselines. Notably, our causal model consistently outperforms all the matching baselines, achieving substantially lower errors across five explained models. Noteworthy, Abraham et al. (2022) proposed the Approx method (the only matching method they examined) and found it outperforms common NLP explainers. Our results show that all matching baselines (including Approx, excluding Random Match and Propensity) are competitive and perform similarly. Nevertheless, they fall short of our causal representation model.

Table 2: Comparison between different methods and baselines. The columns present `Err` scores (Eq. 7) when explaining different five-class sentiment models. The sub-columns present three different measures: Euclidean distance (*L2*), Cosine distance (*Cos*), and norm difference (*ND*). The top table presents scores using a single match ($K = 1$), and the bottom table when $K = 10$. The *Generative* rows are not matching methods and thus are not comparable to such methods. Another non-comparable row is the first, which presents the performance of our causal model with a candidate set that also includes ground-truth (GT) CFs. **Numbers are means over 24 interventions, ↓ is better.**

| $K = 1$ | DistilBERT | | | BERT | | | RoBERTa | | | Llama2-7b | | | Llama2-13b | | | AVG |
| | *L2* | *Cos* | *ND* | *L2* | *Cos* | *ND* | *L2* | *Cos* | *ND* | *L2* | *Cos* | *ND* | *L2* | *Cos* | *ND* | |
|---|---|---|---|---|---|---|---|---|---|---|---|---|---|---|---|---|
| Causal Model (+GT) | **.13** | **.09** | **.06** | **.14** | **.10** | **.07** | **.13** | **.09** | **.06** | **.12** | **.07** | **.06** | **.11** | **.07** | **.06** | **.09** |
| Fine-tune Generative | .38 | .28 | .21 | .42 | .28 | .23 | .39 | .27 | .21 | .36 | .23 | .22 | .31 | .21 | .19 | .28 |
| Few-shot Generative | .42 | .32 | .23 | .45 | .29 | .23 | .41 | .30 | .21 | .38 | .26 | .23 | .33 | .23 | .20 | .30 |
| Zero-shot Generative | .43 | .35 | .24 | .47 | .35 | .25 | .44 | .34 | .23 | .40 | .27 | .25 | .35 | .26 | .21 | .32 |
| Random Match | .88 | .64 | .47 | .96 | .62 | .49 | .95 | .63 | .49 | .84 | .54 | .50 | .84 | .55 | .49 | .66 |
| Propensity | .90 | .68 | .48 | .99 | .69 | .56 | .99 | .68 | .56 | .88 | .61 | .53 | .86 | .58 | .52 | .70 |
| Approx | .74 | .60 | .41 | .81 | .58 | .45 | .79 | .60 | .44 | .71 | .49 | .43 | .70 | .54 | .49 | .58 |
| PT RoBERTa | .75 | .61 | .39 | .83 | .59 | .42 | .81 | .61 | .41 | .74 | .53 | .43 | .73 | .51 | .43 | .59 |
| PT S-Transformer | .75 | .60 | .40 | .83 | .60 | .45 | .81 | .61 | .43 | .74 | .55 | .44 | .69 | .52 | .41 | .59 |
| FT S-Transformer | .72 | .78 | .46 | .82 | .79 | .59 | .80 | .84 | .56 | .73 | .63 | .45 | .68 | .63 | .39 | .66 |
| Causal Model | **.66** | **.55** | **.36** | **.70** | **.55** | **.39** | **.68** | **.56** | **.37** | **.64** | **.47** | **.39** | **.59** | **.45** | **.36** | **.52** |

| $K = 10$ | DistilBERT | | | BERT | | | RoBERTa | | | Llama2-7b | | | Llama2-13b | | | AVG |
| | *L2* | *Cos* | *ND* | *L2* | *Cos* | *ND* | *L2* | *Cos* | *ND* | *L2* | *Cos* | *ND* | *L2* | *Cos* | *ND* | |
|---|---|---|---|---|---|---|---|---|---|---|---|---|---|---|---|---|
| Fine-tune Generative | **.32** | **.21** | **.20** | **.36** | **.23** | **.23** | **.34** | **.21** | **.21** | **.30** | **.18** | **.20** | **.27** | **.16** | **.18** | **.24** |
| Few-shot Generative | .38 | .30 | .22 | .42 | .28 | .24 | .38 | .29 | .21 | .34 | .24 | .22 | .30 | .22 | .19 | .28 |
| Zero-shot Generative | .38 | .32 | .23 | .42 | .31 | .25 | .40 | .31 | .23 | .36 | .24 | .23 | .31 | .22 | .19 | .29 |
| Random Match | .70 | .52 | .40 | .78 | .51 | .47 | .77 | .51 | .45 | .67 | .45 | .40 | .67 | .45 | .40 | .54 |
| Propensity | .72 | .54 | .43 | .80 | .52 | .51 | .79 | .52 | .50 | .68 | .47 | .42 | .69 | .47 | .43 | .57 |
| Approx | .61 | .49 | .37 | .68 | .48 | .43 | .66 | .48 | .41 | .58 | .40 | .37 | .57 | .41 | .36 | .49 |
| PT RoBERTa | .64 | .50 | .36 | .71 | .49 | .44 | .70 | .49 | .42 | .61 | .44 | .37 | .60 | .43 | .36 | .51 |
| PT S-Transformer | .64 | .51 | .38 | .72 | .50 | .45 | .71 | .50 | .44 | .61 | .45 | .38 | .59 | .44 | .36 | .51 |
| FT S-Transformer | .61 | .56 | .43 | .69 | .56 | .50 | .67 | .56 | .47 | .60 | .48 | .41 | .56 | .50 | .36 | .53 |
| Causal Model | **.56** | **.47** | **.35** | **.63** | **.46** | **.41** | **.60** | **.45** | **.38** | **.55** | **.40** | **.36** | **.52** | **.38** | **.33** | **.46** |

Although the generative approach exhibits better scores than the matching approach, it is important to note that the quality of the $\widehat{\text{ICaCE}}_f$ estimation of any matching method is highly dependent on the matching set. Therefore, we also include the `Err` scores for our causal model when the candidate set contains the ground-truth human-written CFs (see the first row in Table 2). In that case, our causal model outperforms even the generative models. This suggests that our method is potentially the SOTA explainer (although this is impractical since access to CFs is a strong assumption).

Finally, Figure 2, which plots the `Err` score when selecting the $k^{\text{th}}$ match, $m_k(x)$, reveals three desirable properties of our causal model: (1) The lowest `Err` score belongs to the first match, indicating that, on average, our causal model indeed selects the best match; (2) A monotonically increasing pattern (i.e., a larger $k$ results in a higher `Err`) demonstrates a strong association between representation similarity and explainability capacity: The more similar a match is, the more accurate the $\widehat{\text{ICaCE}}_f$ estimator becomes; and (3) The top-ranked matches selected by our causal model outperform other baselines, achieving notably lower `Err` scores.

**3. Top-$K$ matching improves the explanation capabilities of every method.** When comparing the top rows of Table 2 ($k = 1$) to the bottom rows ($k = 10$), it is easy to observe that Top-$K$ matching lowers the `Err` of every examined explanation method. This observation also holds for the generative models, which accordingly generate multiple CFs via sampling. The definitive results, combined with the simplicity and cost-effectiveness of the Top-$K$ extension (see Appendix §B), suggest that top-$K$ should be a standard for explainers. The question that arises is *what should the value of K be*?

In Figure 3, we plot the L2 `Err` as a function of $k$ for our method and three other baselines when explaining DistilBERT (the trends are similar for the other metrics and explained models). The prominent '✓' shape curve of the causal model is desirable and indicates that the model selects good matches at first. Then, around $k = 20$, the error begins to surge and only after $k = 100$ that it matches the error at $k = 1$. Conversely, other baselines rank candidates less effectively and exhibit a more

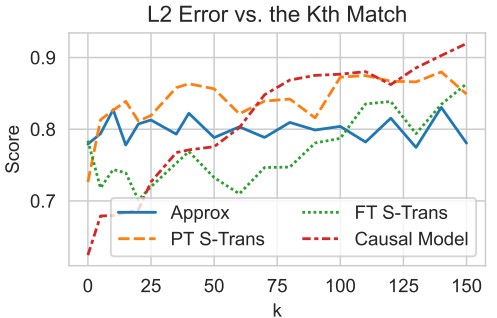
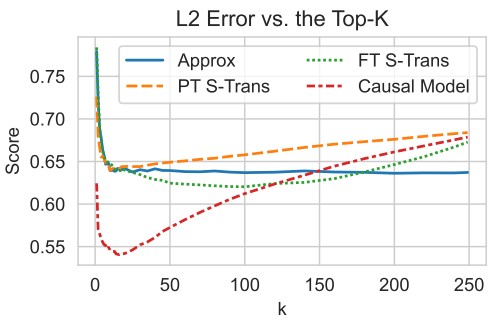

Figure 2: The L2 `Err` score for explaining the DistilBERT model (Y-axis) as a function of selecting the $k^{\text{th}}$ match, $m_k(x)$. For Figures 2 and 3, we present average `Err` scores only for interventions with a candidate set of size larger than 250, making the numbers differ from Table 2.

Figure 3: The L2 `Err` score for explaining the DistilBERT model (Y-axis) as a function of Top-$K$ matching. The figure illustrates the beneficial impact of considering multiple matches. However, beyond a certain point, adding more matches negatively affects causal estimation.

gradual increase after their minimum. Nonetheless, the Top-$K$ approach has a beneficial effect for all methods and a broad range of values can be picked before the effect fades. We arbitrarily opted for $k = 10$ in this study, even though $k = 20$ would have been more advantageous for our causal model. A future direction could explore how to tune the $k$ parameter for each individual example.

## 6 CONSTRUCTING A NEW BENCHMARK FOR MODEL EXPLANATION

While we aim to validate our earlier conclusions across multiple benchmarks, CEBaB remains the sole high-quality benchmark for NLP explainers. On the other hand, we show that LLMs generate CFs that resemble human-written CFs. Consequently, we harness LLM capabilities to emulate CEBaB and propose a new benchmark for model explanation, eliminating the high cost of manually annotating such benchmarks. **In Appendix §D, we thoroughly detail the new benchmark, including the construction process and the new experiments we conduct.** We decided to focus on Stance Detection, which is an important and well-studied NLP task that determines the position (support, oppose, neutral) expressed by the speaker towards a particular topic (e.g., abortions, climate change). We rely on the stance detection dataset from the SemEval 2016 shared task (Mohammad et al., 2016).

For simplicity of the new setup, we embrace the same causal graph of CEBaB (see Figure 1), but replace the four review concepts with new relevant concepts: (1) Tweet's subject (Abortions, Atheism, Climate Change, Feminism, Hillary Clinton); (2) Speaker's age (Unknown, Teenager, Elder); (3) Speaker's gender (Unknown, Female, Male); and (4) Speaker's Job (Unknown, Farmer, Professor). Using GPT-4, we generate new tweets and golden CFs for the test set, enabling the evaluation of explanation methods. Furthermore, unlike CEBaB, this dataset allows evaluation in an out-of-distribution setting, such as when the matching candidate set compromises texts discussing subjects differing from the subject of the test examples. We then test our methods under distribution shift, validating the robustness of our main results from the original CEBaB dataset (results in Table 6).

## 7 DISCUSSION

In this paper, we explore the terrain of explaining the impact of real-world concepts on the predictions of NLP models. We focus on model-agnostic techniques, which do not require access to the explained model during the training time and hence can explain numerous models. We hope this paper may direct the attention of the NLP community toward the vital intersection of causal inference and model interpretability. We further motivate the utilization of LLM for a model explanation, either for generating CFs or for guiding efficient approximation methods. Nevertheless, the gap that still exists between the two approaches demonstrates the need for further research. In Appendix §F we discuss the requirements, limitations, and the applicability of our proposed methods to complex causal graphs. In Appendix §E, we extend this discussion and emphasize the key implications of our findings.

**Ethics statement.** While explanation methods offer deeper transparency of NLP models, researchers and practitioners should remain vigilant and mindful. These methods, if misused, could over-emphasize or amplify biases present in the model or data. Our new dataset (§6), derived from tweets taken from SemEval16 (Mohammad et al., 2016), could inherently contain societal biases, including, but not limited to, social prejudices or racism. We acknowledge that tweets can sometimes reflect or amplify societal sentiments, both positive and negative. Furthermore, by leveraging GPT-4 to edit these tweets (e.g., changing the gender of the writer), we recognize the potential for introducing additional biases.

**Reproducibility.** We have made comprehensive efforts to ensure that researchers can effectively reproduce the results presented in our paper: *(1) Theoretical Foundations:* All pertinent theoretical results, encompassing definitions, assumptions, and proofs, are comprehensively detailed in Appendix A. *(2) Prompts:* The specific prompts we employed for generating examples and counterfactuals are laid out in Appendix I. *(3) Code:* The code is available in `https://github.com/YairGat/causal-explanations`. The hyperparameters we used are described in G.

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

# Appendix

## Table of Contents

## A A FAITHFUL EXPLANATION

In this section, we prove our main theorem from §3.1. We start by providing definitions for explanation methods and then continue to define the order-faithfulness property. For simplicity, assume the image of the explained models is $\mathbb{R}$.

**Definition 1** (Approximated Counterfactual). *Given a DGP $\mathcal{G}$, an explained model $f$, an example $\boldsymbol{x}$ whose treatment value is $T(\boldsymbol{x}) = t$, and an intervention $T : t \to t'$, we call $\tilde{\boldsymbol{x}}_{t'}$ **approximated counterfactual** if:*

$$f(\tilde{\boldsymbol{x}}_{t'}) = \begin{cases} f(\boldsymbol{x}) & t' = t \\ f(\boldsymbol{x}_{T=t'}) + \epsilon & otherwise \end{cases} \qquad \mathbb{E}[\epsilon] = 0$$

*Where $\boldsymbol{x}_{T=t'}$ is the golden CF of $\boldsymbol{x}$ in $\mathcal{G}$ and $\epsilon$ is an approximation error.*

This definition also suggests why the Top-$K$ technique improves the causal effect estimation: Averaging the prediction of $K$ approximated CFs reduces the variance of the approximation error, making the estimator more robust.

**Definition 2** (Approximated Counterfactual Explanation Method). *Given a dataset $\mathbb{D}$ sampled from a DGP $\mathcal{G}$, an explained model $f$ and an intervention $T : t \to t'$, the **approximated counterfactual explanation method** $S_{CF}$ is defined to be:*

$$S_{CF}(f, T, t, t') = \frac{1}{|\mathbb{D}|} \sum_{x \in \mathbb{D}} f(\tilde{\boldsymbol{x}}_{t'}) - f(\tilde{\boldsymbol{x}}_t)$$

**Definition 3** (Non-causal Explanation Method). *Given a dataset $\mathbb{D}$ sampled from a DGP $\mathcal{G}$, an explained model $f$ and an intervention $T : t \to t'$, let $D[X, T, f]$ be defined to be a set that contains for each unit from $\mathbb{D}$ a triplet of model input, model output, and treatment assignment:*

$$D[X, f, T] = \{(x(\boldsymbol{u}), f(x(\boldsymbol{u}), T(\boldsymbol{u}))) \,|\, \boldsymbol{u} \in \mathbb{D}\}$$

*An explanation method is called **non-causal explanation method** $S_{NC}$ if it is a function of $D[X, f, T]$:*

$$S_{NC}(f, T, t, t') = h\left(D[X, f, T]\right)$$

For simplicity, we notate the triplet $(x(\boldsymbol{u}), f(x(\boldsymbol{u}), T(\boldsymbol{u})))$ with $(\boldsymbol{x}, f(\boldsymbol{x}), T(\boldsymbol{x}))$. According to Def. 3, the training data of $S_{NC}$ is $D[X, f, T]$, meaning it can be, for example, an unbiased estimator of: $\mathbb{E}_{\mathcal{G}}[f(x)|T = t'] - \mathbb{E}_{\mathcal{G}}[f(x)|T = t]$. However, it may also overlook other concepts than the treatment, such as confounding concepts, which may hint why $S_{NC}$ potentially fails.

**Definition 4** (Order-Faithfulness). *Given an i.i.d. dataset $\mathbb{D}$ sampled from a DGP $\mathcal{G}$, an explained model $f$ and two interventions $C_1 : c_1 \to c_1', C_2 : c_2 \to c_2'$, an explanation method $S$ is **order-faithful** if:*

$$\mathbb{E}_{\mathbb{D}\sim\mathcal{G}}[S(f, C_1, c_1, c_1')] > \mathbb{E}_{\mathbb{D}\sim\mathcal{G}}[S(f, C_2, c_2, c_2')] \iff \mathsf{CaCE}_f(C_1, c_1, c_1') > \mathsf{CaCE}_f(C_2, c_2, c_2') \quad (8)$$

**Lemma 1.** *For an explained model $f$, $S_{CF}$ is order-faithful for any DGP $\mathcal{G}$ and any interventions.*

*Proof.* We start by connecting the approximated CF to the do operator. From Def. 1, it follows that the expected prediction of an approximated CF is equal to the interventional one:

$$\mathbb{E}_{\boldsymbol{x}\sim\mathcal{G}}[f(\tilde{\boldsymbol{x}}_{t'})] = \mathbb{E}_{\boldsymbol{x}\sim\mathcal{G}}[f(\boldsymbol{x}_{T=t'}) + \epsilon] = \mathbb{E}_{\boldsymbol{x}\sim\mathcal{G}}[f(\boldsymbol{x}_{T=t'})] = \mathbb{E}_{\boldsymbol{x}\sim\mathcal{G}}\left[f(\boldsymbol{x})|do(T = t')\right]$$

Combining this result with the Def. 2 and the fact that $P_{\mathbb{D}} = P_{\mathcal{G}}$, we get:

$$\begin{aligned} \mathbb{E}_{\mathbb{D}\sim\mathcal{G}}[S_{CF}(f, T, t, t')] &= \mathbb{E}_{\mathbb{D}\sim\mathcal{G}}\left[\frac{1}{|\mathbb{D}|} \sum_{x \in \mathbb{D}} f(\tilde{\boldsymbol{x}}_{t'}) - f(\tilde{\boldsymbol{x}}_t)\right] \\ &= \mathbb{E}_{x\sim\mathbb{G}}[f(\tilde{\boldsymbol{x}}_{t'}) - f(\tilde{\boldsymbol{x}}_t)] \\ &= \mathbb{E}_{x\sim\mathcal{G}}[f(\tilde{\boldsymbol{x}}_{t'})] - \mathbb{E}_{x\sim\mathcal{G}}[f(\tilde{\boldsymbol{x}}_t)] \\ &= \mathbb{E}_{\boldsymbol{x}\sim\mathcal{G}}\left[f(\boldsymbol{x})|do(T = t')\right] - \mathbb{E}_{\boldsymbol{x}\sim\mathcal{G}}\left[f(\boldsymbol{x})|do(T = t)\right] \\ &= \mathsf{CaCE}_f(T, t, t') \end{aligned}$$

Thus, $S_{CF}$ is an unbiased estimator of $\mathsf{CaCE}_f$, and it holds that:

$$\mathbb{E}_{\mathbb{D}\sim\mathcal{G}}[S_{CF}(f, C_1, c_1, c_1')] - \mathbb{E}_{\mathbb{D}\sim\mathcal{G}}[S_{CF}(f, C_2, c_2, c_2')] = \mathsf{CaCE}_f(C_1, c_1, c_1') - \mathsf{CaCE}_f(C_2, c_2, c_2')$$

Meaning that $S_{CF}$ is order-faithful for any $\mathcal{G}$ and any interventions. □

**Theorem 1.** *For an explained model $f$, (1) The approximated CF explanation method $S_{CF}$ is order-faithful for every DGP $\mathcal{G}$ and two interventions; (2) For every DGP $\mathcal{G}$, there exist a DGP $\mathcal{G}'$ resulting from a small modification of $\mathcal{G}$, a model $f'$, and two interventions $C_1 : c_1 \rightarrow c_1', C_2 : c_2 \rightarrow c_2'$ such that the non-causal explanation method $S_{NC}$ is not order-faithful.*

*Proof.* From Lemma 1 we know that $S_{CF}$ is order-faithful for any DGP and (1) holds.

For part (2) of the theorem, let $\mathcal{G}$ be a DGP with at least two concepts $C_1, ..., C_K$ (in case $\mathcal{G}$ has only one concept, we can add concept to the DGP and continue with the proof). If $S_{NC}$ is not order-faithful for some two interventions, set $f' = f$ and we are done. Therefore, we assume that $S_{NC}$ is order-faithful for all interventions in $\mathcal{G}$ and w.l.o.g. assume that:

(a) $C_1$ has no parent nodes (except exogenous variables). It is guaranteed that at least one concept has no parents since the DGP is a DAG.

(b) The causal effect of $C_1$ is bigger than of $C_2$: $\mathsf{CaCE}_f(C_1, c_1, c_1') > \mathsf{CaCE}_f(C_2, c_2, c_2')$. Otherwise, inverse the corresponding signs of the proof.

Proof sketch of the second part of the theorem: We will construct a new DGP $\mathcal{G}'$ by introducing a new unobserved confounding concept $C_0$ (this is the "small modification of $\mathcal{G}$" from the theorem) and a new explained model $f'$, such that:

$$\forall i \geq 1 : P_{\mathcal{G}'}(\boldsymbol{x}, f'(\boldsymbol{x}), C_i) = P_{\mathcal{G}}(\boldsymbol{x}, f(\boldsymbol{x}), C_i) \tag{9}$$

$$\mathsf{CaCE}_{f'}(C_1, c_1, c_1') < \mathsf{CaCE}_{f'}(C_2, c_2, c_2') \tag{10}$$

Notice that the first condition Eq. 9 ensures that $S_{NC}$ produces the same explanations also in the new DGP. This is because for an intervention of a concept $C_i$, $S_{NC}$ is a function of $D[X, f, C_i]$, and the joint distribution from which it is sampled, $P_{\mathcal{G}'}(\boldsymbol{x}, f'(\boldsymbol{x}), C_i)$, is the same as in the original DGP $\mathcal{G}$.

Moreover, according to the second condition Eq. 10, for the new $\mathcal{G}'$ and $f'$, the causal effect of $C_1$ and $C_2$ is reversed (compared to the original $\mathcal{G}$ and $f$). However, $S_{NC}$ produces the same explanations as before (for $\mathcal{G}$ and $f$), meaning it cannot be order-faithful also for $\mathcal{G}'$ and $f'$.

We next show that such $\mathcal{G}'$ and $f'$ exist.

**Construction:** Let $f'$ be a new explained model and $\mathcal{G}'$ be a copy of $\mathcal{G}$ with a new node $C_0$, and new edges $(C_0, C_1), (C_0, f'(X))$ (i.e., $C_0$ is a confounder). $C_0$ accepts three values: 0, 1 and 2. Let $p = P_{\mathcal{G}}(C_1 = c_1)$ and $p' = P_{\mathcal{G}}(C_1 = c_1')$. Define the following probabilities:

$$P_{\mathcal{G}'}(C_0 = 0) = p$$
$$P_{\mathcal{G}'}(C_0 = 1) = p'$$
$$P_{\mathcal{G}'}(C_0 = 2) = 1 - p - p'$$
$$P_{\mathcal{G}'}(C_1 = c_1 | C_0 = 0) = P_{\mathcal{G}'}(C_1 = c_1' | C_0 = 1) = 1$$
$$P_{\mathcal{G}'}(C_1 = c_1 | C_0 = 2) = P_{\mathcal{G}'}(C_1 = c_1' | C_0 = 2) = 0$$
$$\forall c \neq c_1, c_1' : P_{\mathcal{G}'}(C_1 = c | C_0 = 2) = \frac{P_{\mathcal{G}}(C_1 = c)}{1 - p - p'}$$

We conclude that the marginal distribution of $C_1$ is equal in both DGPs, $P_{\mathcal{G}'}(C_1) = P_{\mathcal{G}}(C_1)$:

$$P_{\mathcal{G}'}(C_1 = c_1) = P_{\mathcal{G}'}(C_1 = c_1 | C_0 = 0) P_{\mathcal{G}'}(C_0 = 0) = p \cdot 1 = P_{\mathcal{G}}(C_1 = c_1)$$
$$P_{\mathcal{G}'}(C_1 = c_1') = P_{\mathcal{G}'}(C_1 = c_1' | C_0 = 1) P_{\mathcal{G}'}(C_0 = 1) = p' \cdot 1 = P_{\mathcal{G}}(C_1 = c_1')$$
$$\forall c \neq c_1, c_1' : P_{\mathcal{G}'}(C_1 = c) = P_{\mathcal{G}'}(C_1 = c | C_0 = 2) P_{\mathcal{G}'}(C_0 = 2)$$
$$= (1 - p - p') \cdot \frac{P_{\mathcal{G}}(C_1 = c)}{1 - p - p'} = P_{\mathcal{G}}(C_1 = c)$$

Let $d$ be a difference in causal effects between $C_1$ and $C_2$ for $f$, which is positive following assumption (b) from the beginning of the proof.

$$d = \mathsf{CaCE}_f(C_1, c_1, c_1') - \mathsf{CaCE}_f(C_2, c_2, c_2') > 0$$

Let $f'$ be the new explained model. $f'$ has an oracle access[1] to the underlying concepts and is defined to be:
$$f'(\boldsymbol{x}) = f(\boldsymbol{x}) - 2d \cdot \mathbb{I}_{[C_1(\boldsymbol{x})=c_1]} + 2d \cdot \mathbb{I}_{[C_0(\boldsymbol{x})=0]}$$
Where $\mathbb{I}$ is the indicator function. If $C_0(\boldsymbol{x}) = 0$ then $C_1(\boldsymbol{x}) = c_1$ with probability 1 and:
$$f'(\boldsymbol{x}) = f(\boldsymbol{x}) - 2d \cdot \mathbb{I}_{[True]} + 2d \cdot \mathbb{I}_{[True]} = f(\boldsymbol{x}).$$
Conversely, if $C_0(\boldsymbol{x}) = 1$ then $C_1(\boldsymbol{x}) = c_1'$ with probability 1 and:
$$f'(\boldsymbol{x}) = f(\boldsymbol{x}) - 2d \cdot \mathbb{I}_{[False]} + 2d \cdot \mathbb{I}_{[False]} = f(\boldsymbol{x})$$
Finally, if $C_0(\boldsymbol{x}) = 2$ then $C_1(\boldsymbol{x}) \neq c_1, c_1'$ and:
$$f'(\boldsymbol{x}) = f(\boldsymbol{x}) - 2d \cdot \mathbb{I}_{[False]} + 2d \cdot \mathbb{I}_{[False]} = f(\boldsymbol{x})$$
From the above equations, it is clear that in the DGP $\mathcal{G}'$ without any interventions, for all values of $C_0$ given $\boldsymbol{x}$ we have $f'(\boldsymbol{x}) = f(\boldsymbol{x})$. Meaning that:
$$P_{\mathcal{G}'}(f'(X)|C_0, X) = P_{\mathcal{G}'}(f'(X)|X) = P_{\mathcal{G}}(f(X)|X)$$

We have shown that $P_{\mathcal{G}'}(C_1) = P_{\mathcal{G}}(C_1)$. Since $\mathcal{G}'$ is a copy of $\mathcal{G}$ (except $C_0$), and since neither the marginal distribution of the concepts nor the generation process of $X$ did not change, then:
$$\forall i = 1, ..., K : P_{\mathcal{G}}(C_i) = P_{\mathcal{G}'}(C_i)$$
$$\forall i = 1, ..., K : P_{\mathcal{G}}(X|C_i) = P_{\mathcal{G}'}(X|C_i)$$
Finally, from definition of $f$, it depends only on $\boldsymbol{x}$, thus: $\forall i : P_{\mathcal{G}}(f(X)|X, C_i) = P_{\mathcal{G}}(f(X)|X)$ (the same holds for $f'$, and we have shown above that $P_{\mathcal{G}'}(f'(X)|C_0, X) = P_{\mathcal{G}'}(f'(X)|X)$). Therefore:
$$\begin{aligned}\forall i = 1, ..., K : P_{\mathcal{G}}&(X, f(X), C_i) \\ &= P_{\mathcal{G}}(C_i)P_{\mathcal{G}}(X|C_i)P_{\mathcal{G}}(f(X)|X, C_i) \\ &= P_{\mathcal{G}}(C_i)P_{\mathcal{G}}(X|C_i)P_{\mathcal{G}}(f(X)|X) \\ &= P_{\mathcal{G}'}(C_i)P_{\mathcal{G}'}(X|C_i)P_{\mathcal{G}'}(f'(X)|X) \\ &= P_{\mathcal{G}'}(C_i)P_{\mathcal{G}'}(X|C_i)P_{\mathcal{G}'}(f'(X)|X, C_i) \\ &= P_{\mathcal{G}'}(X, f'(X), C_i)\end{aligned}$$
Eq. 9 holds and accordingly, $P_{\mathcal{G}'}(D[X, f, C_i]) = P_{\mathcal{G}}(D[X, f, C_i])$, meaning that $S_{NC}$ produces the same explanations for $\mathcal{G}$ and $\mathcal{G}'$. Since $S_{NC}$ is order-faithful in $\mathcal{G}$, and we know from our assumptions on $\mathcal{G}$ that $\mathsf{CaCE}_f(C_1, c_1, c_1') > \mathsf{CaCE}_f(C_2, c_2, c_2')$, in follows then that:
$$\mathbb{E}_{\mathbb{D}\sim\mathcal{G}'}[S(f', C_1, c_1, c_1')] > \mathbb{E}_{\mathbb{D}\sim\mathcal{G}'}[S(f', C_2, c_2, c_2')]$$

We next show that Eq. 10 holds (reversed causal effect), thus $S_{NC}$ cannot be order-faithful in $\mathcal{G}'$:
$$\begin{aligned}\mathsf{CaCE}_{f'}&(C_1, c_1, c_1') \\ &= \mathsf{CaCE}_f(C_1, c_1, c_1') - 2d \cdot \mathsf{CaCE}_{\mathbb{I}_{[C_1(x)=c_1]}}(C_1, c_1, c_1') + 2d \cdot \mathsf{CaCE}_{\mathbb{I}_{[C_0(x)=1]}} \\ &= \mathsf{CaCE}_f(C_1, c_1, c_1') - 2d \\ &= \mathsf{CaCE}_f(C_1, c_1, c_1') - (\mathsf{CaCE}_f(C_1, c_1, c_1') - \mathsf{CaCE}_f(C_2, c_2, c_2')) - d \\ &= \mathsf{CaCE}_f(C_2, c_2, c_2') - d < \mathsf{CaCE}_f(C_2, c_2, c_2') = \mathsf{CaCE}_{f'}(C_2, c_2, c_2')\end{aligned}$$

The last equation is true as $f'$ is not affected by $C_2$ in any way except through the $f$.

$\square$

---

[1] In practice, it can be that the unobserved variable $C_0$ controls for some modification $\psi(\boldsymbol{x})$ and the path $X \to f(X)$ is replaced in the new DGP $\mathcal{G}'$ by $X \to \psi(X) \to f'(X)$ and $C_0 \to \psi(X)$. Nevertheless, $S_{NC}$ has only access to **realizations** of $X$ and $f'$, and therefore, for simplicity of the notations we assume an oracle.

Table 3: Comparison between the inference time latency (in seconds) of different methods. The top rows present counterfactual generation models, and the middle and bottom tables for matching methods. Notice that all the baselines described in §4.2 have the same latency as the Causal Model and thus are not specified.

| Method | Backbone | Explaining a single example | | | Explaining 100 examples | | |
|---|---|---|---|---|---|---|---|
| | | $Top\text{-}K = 1$ | $Top\text{-}K = 10$ | $Top\text{-}K = 100$ | $Top\text{-}K = 1$ | $Top\text{-}K = 10$ | $Top\text{-}K = 100$ |
| Fine-tune Generative | T5-Base | 0.84 | 1.03 | 3.09 | 84 | 102.2 | 308.2 |
| Zero-shot Generative | ChatGPT (turbo) | 2.45 | 2.95 | 4.47 | 245.3 | 295.3 | 447.2 |
| Few-shot Generative | ChatGPT (turbo) | 2.52 | 2.98 | 4.49 | 252.3 | 298.3 | 449.2 |
| **250 candidates** | | $Top\text{-}K = 1$ | $Top\text{-}K = 10$ | $Top\text{-}K = 100$ | $Top\text{-}K = 1$ | $Top\text{-}K = 10$ | $Top\text{-}K = 100$ |
| Approx | – | 0.86 | 0.86 | 0.86 | 1.95 | 1.95 | 1.95 |
| Causal Model | S-Transformer | 0.03 | 0.03 | 0.03 | 0.27 | 0.27 | 0.27 |
| **1000 candidates** | | $Top\text{-}K = 1$ | $Top\text{-}K = 10$ | $Top\text{-}K = 100$ | $Top\text{-}K = 1$ | $Top\text{-}K = 10$ | $Top\text{-}K = 100$ |
| Approx | – | 0.86 | 0.86 | 0.86 | 1.95 | 1.95 | 1.95 |
| Causal Model | S-Transformer | 0.03 | 0.03 | 0.03 | 0.27 | 0.27 | 0.27 |

## B INFERENCE TIME EFFICIENCY

As discussed in § 1, we seek three attributes in the explanation method: It should be model-agnostic, effective, and efficient. In this subsection, we focus on inference efficiency, an aspect that is vital for providing real-time explanations and explaining vast amounts of data.

To compare the computational efficiency of various explanation methods, we present the inference time latency in Table 3. This latency represents the time (in seconds) required to compute the $\widehat{\text{ICaCE}}_f$ for either a single example or a batch of 100 examples. We measure this latency for Top-$K = 1, 10, 100$ using the following methods: (1) Generative baselines; (2) The Approx baseline, which employs three fine-tuned RoBERTa models. We first use these models to predict the three confounder concepts of the query examples. Following this, we randomly select matches that correspond to the values of each query example; and (3) Our causal matching method, which first represents the query examples, then computes their cosine similarity with the candidates and finally finds the most similar matches. Notice that other matching baselines have the same latency as our method. All the query and candidate examples are 50 tokens in length, which is also the length of the model-generated CFs. For the matching baselines, we utilized a candidate set of 1000 examples.

As can be seen in Table 2, the latency of the explanation methods we evaluated varies dramatically between them. Interestingly, Top-$K$ does not influence the latency of the matching methods since it merely involves an argmax operation. Similarly, the size of the candidate set has no noticeable impact on latency. Our method also outpaces the Approx baseline, as it utilizes fewer models and avoids the need to filter the candidate set based on the values of the confounder concepts.

The generative baselines are significantly slower than the matching methods, primarily due to the autoregressive token-by-token generation process. For instance, when explaining a single query example, the fine-tuned model is around 30 times slower than matching methods for $K = 1$ and 100 times slower for $K = 100$. For batch processing (explaining 100 queries), the benefits of parallelism offered by GPUs make matching methods exceptionally fast. Our method is 1000 times faster than the generative baselines.

## C ABLATION STUDY

This subsection aims to shed more light on the effect of each of our design choices. To this end, we examine variations or omissions of components from our causal representation model. Noteworthy, with the original candidate set, which is relatively small, all of the ablation models are competitive, and the performance difference is insignificant. Therefore, we also experiment with variations of the candidate set that reveal the pitfalls of each ablation model. Table 4 presents the performance when using three candidate sets: (1) The original candidate set, containing only original reviews from the matching split; (2) The original set augmented by the ground-truth human-written CFs; and (3) The original set augmented by the misspecified model-generated CFs.

We start by examining the *backbone encoder model* of the causal model: SentenceTransformer and RoBERTa. Comparing the first two rows of Table 4, we observe that both encoders are competitive and can be utilized as the backbone. We decided to use SentenceTransformer because its development

Table 4: Ablation study. Row 1: Our model. Row 2: With a different backbone. Row 3: Causal S-Transformer (ours) without filtering misspecified CFs from $\mathbb{X}_{\text{CF}}$. Row 4: Without human-annotated concepts (i.e., concepts are predicted by a zero-shot LLM). Rows 5-10: Different variations of the objective (Eq. 5) when discarding components that include the specified sets. Columns present different matching candidate sets: original, original with ground-truth CFs, and original with misspecified CFs. **Numbers are the mean over 5 explained models and 24 interventions.** ↓ is better.

|  |  | Original | | | + GT CFs | | | + Miss. CFs | | |
|---|---|---|---|---|---|---|---|---|---|---|
|  |  | *L2* | *Cos* | *norm* | *L2* | *Cos* | *norm* | *L2* | *Cos* | *norm* |
| 1 | Causal S-Trans. | .65 | .52 | .38 | .13 | .09 | .06 | .70 | .58 | .46 |
| 2 | Causal Roberta | .66 | .52 | .37 | .16 | .11 | .08 | .70 | .59 | .46 |
| 3 | w/o filtering | .66 | .51 | .38 | .12 | .08 | .05 | .77 | .63 | .52 |
| 4 | w/o labels | .65 | .52 | .36 | .13 | .09 | .06 | .67 | .59 | .43 |
| 5 | w/o $\mathbb{X}_{\text{CF}}$ | .67 | .54 | .39 | .26 | .18 | .13 | .69 | .58 | .45 |
| 6 | w/o $\mathbb{X}_{\text{M}}$ | .66 | .51 | .37 | .09 | .06 | .04 | .81 | .68 | .57 |
| 7 | w/o $\mathbb{X}_{\neg\text{CF}}$ | .66 | .51 | .38 | .06 | .04 | .03 | .83 | .68 | .60 |
| 8 | w/o $\mathbb{X}_{\neg\text{M}}$ | .66 | .51 | .38 | .16 | .11 | .07 | .70 | .59 | .46 |
| 9 | w/o $\mathbb{X}_{\text{M}} \cup \mathbb{X}_{\neg\text{M}}$ | .66 | .53 | .39 | .19 | .13 | .09 | .74 | .61 | .50 |
| 10 | w/o $\mathbb{X}_{\text{CF}} \cup \mathbb{X}_{\neg\text{CF}}$ | .66 | .52 | .38 | .15 | .09 | .07 | .72 | .60 | .49 |

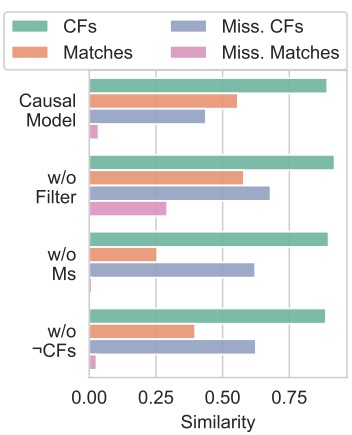

Figure 4: Average similarity between query examples and examples from the four sets (the bars), plotted for different variants of the causal model (Y-axis).

loss was lower. By comparing the first row to the third, we examine the impact *filtering unsuccessful or misspecified CFs* from $\mathbb{X}_{\text{CF}}$. Without filtering, the causal model prioritizes misspecified CFs above valid matches (see Figure 4), and as a result, when misspecified CFs are present in the candidate set, the performance degrades (third row, right column, gray cells).

In row 4, we investigate the applicability of our method without human-annotated concepts in the training dataset (i.e., completely unsupervised setup). Utilizing an LLM (ChatGPT), we predicted concept values in a zero-shot manner. Row 4 reveals that the performance remains consistent and on par with models trained on human-annotated datasets. Surprisingly, it performs better in some metrics (although insignificant), likely because the LLM predicts annotations that are sometimes missed due to disagreements between annotators.

We next examine the impact of *discarding components from the model objective Eq 5*. To this end, for each set of examples, $\mathbb{X}_{\text{CF}}$, $\mathbb{X}_{\text{M}}$, $\mathbb{X}_{\neg\text{CF}}$, and $\mathbb{X}_{\neg\text{M}}$, we train an encoder model while ignoring the three components from the objective that utilize examples from the discarded set. In addition, we also train two encoder models using only model-generated CFs (w/o $\mathbb{X}_{\text{M}}$ and $\mathbb{X}_{\neg\text{M}}$), and without any (w/o $\mathbb{X}_{\text{CF}}$ and $\mathbb{X}_{\neg\text{CF}}$, i.e., training only with human-written reviews – not CFs).

The behavior of each ablation model is as one would expect. For example, when $\mathbb{X}_{\text{CF}}$ is discarded from the objective (row 5), the ablation model struggles to identify CFs and the performance declines when the matching sets include ground-truth CFs. Conversely, when $\mathbb{X}_{\neg\text{CF}}$ or $\mathbb{X}_{\text{M}}$ are excluded (rows 7, 8), the model fails to distinguish between CFs and misspecified ones, often favoring the latter over valid matches. Consequently, the performance drops when the candidate set contains misspecified CFs. *Only our causal model consistently performs well across both candidate sets.* Notably, we can see that even without model-generated CFs (row 10), the causal model performs well. Although the causal estimation is less precise, *achieving a good explainer without utilizing an LLM is possible.*

Finally, Figure 4 presents the average similarity between query examples and the four sets for the causal and selected ablative models. As can be seen, our causal model learns the desirable ranking order: $\boldsymbol{x}_{\neg\text{M}} \preceq \boldsymbol{x}_{\neg\text{CF}} \preceq \boldsymbol{x}_{\text{M}} \preceq \boldsymbol{x}_{\text{CF}}$. In contrast, the model learns to favor misspecified CFs over matches when discarding $\mathbb{X}_{\text{M}}$ or $\mathbb{X}_{\neg\text{CF}}$ or when not performing filtering of misspecified CFs from $\mathbb{X}_{\text{CF}}$. Although these ablation models have a competitive performance to our model and some outperform it when the candidate set contains ground-truth CFs, they fail when the set contains misspecified CFs, highlighting the importance of learning the right ranking order.

Table 5: Examples from the new Stance Detection setup. In the first stage of the dataset construction (top table), we split the original tweets into the train, dev, matching, and test sets. In the second stage (middle table), we keep 30% of the original tweets, and for the remaining 70%, we randomly assign values for the three writer's concepts. In this example, we ask GPT-4 to edit the tweet "to make it sound as if a teenage female wrote it". In the third (bottom table), we generate CFs for the test set (and are used for benchmarking). In this example, we ask GPT-4 to change the Job concept to Farmer.

| | | | | |
|---|---|---|---|---|
| **S1: Original tweet.** I need feminism because "what were you wearing" shouldn't be a question when I tell my stepmom I was cat called. #SemST | | | | |
| $Subject$ : Feminism | $Age$ : ? | $Gender$ : ? | $Job$ : ? | $Label$ : Favor |
| **S2: Modified tweet.** OMG, like, I totally need feminism cuz 'what were you wearing?' shouldn't even be a thing when I tell my stepmom some guy shouted stuff at me. #JustSaying | | | | |
| $Subject$ : Feminism | $Age \rightarrow$ Teen | $Gender \rightarrow$ ♀ | $Job$ : ? | $Label$ : Favor |
| **S3: CF (test set).** Gosh, I reckon I'd holler for feminism, cause 'what were ya wearin?' ought not be a question when I tell ma about some fella yelled stuff at me while I was out in the fields. #JustSaying | | | | |
| $Subject$ : Feminism | $Age$ : Teen | $Gender$ : ♀ | $Job \rightarrow$ Farmer | $Label$ : Favor |

## D  NEW SETUP: EXPLAINING STANCE DETECTION MODELS

While we aim to validate our earlier conclusions across multiple benchmarks, CEBaB remains the sole high-quality benchmark for NLP explainers. On the other hand, we show that LLMs generate CFs that resemble human-written CFs. Consequently, we harness LLM capabilities to emulate CEBaB and propose a new benchmark for model explanation, eliminating the high cost of manually annotating such benchmarks. We decided to focus on Stance Detection, which is an important and well-studied NLP task that determines the position (support, oppose, neutral) expressed by the writer towards a particular topic (e.g., abortions, climate change) (Küçük & Can, 2021; Hardalov et al., 2022; Viehmann et al., 2023). This task is intriguing because understanding public opinions and beliefs can guide decision-making, policy formulation, and marketing strategies. Estimating the causal effect of high-level concepts like gender or age may shed light on the societal biases and perceptions that drive social trends.

We rely on the stance detection dataset from the SemEval 2016 shared task (Mohammad et al., 2016). This dataset contains tweets about five subjects: Abortion, atheism, climate change, feminism, and Hillary Clinton. Humans annotated the stance of the tweets with three possible labels: support, oppose, and neutral. For simplicity of the new setup, we embrace the same causal graph of CEBaB (see Figure 1), but replace the four review concepts with new relevant concepts: (1) Tweet's subject (specified above); (2) Writer's age - teenage, elder or unknown; (3) Writer's gender - female, male or unknown; and (4) Writer's Job - farmer, professor or unknown. The four concepts have an exogenous variable $U$ as their parent and are the direct cause of the text $X$ and the label (stance) $Y$, which is naturally affected by the four concepts. The explanation methods should estimate the effect of 9 interventions: changing the value of the three writer's concepts (age, gender, job) to three values (we do not explain the tweet's subject concept).

**Constructing the new setup.** We follow three steps for constructing the new benchmark. See the examples in Table 5. In the first stage, we randomly split the data into four sets: train ($N = 1000$), matching ($N = 2250$), dev ($N = 250$), and test ($N = 500$). Each subject is precisely one-fifth of each split. For each set and subject, we keep 30% of the original tweet and assign an 'unknown' value to the writer's concepts. For the remaining 70%, we randomly sample two of the three writer's concepts and assign new values (e.g., male, unknown age, professor). Noteworthy, we do not uniformly sample the writer's profile since we want to introduce correlations between the concepts and the label. Table 7 presents the joint probabilities of different concept and label values.

In the second stage, we utilize GPT-4 to generate new tweets according to the new value assignment of 70% of the tweets. The prompts we used are described in Figure 10. In the third and final stage, we utilize GPT-4 to generate the ground-truth CFs for the test set, which enables calculating the Err for evaluating explanation methods. Accordingly, for each test example, we prompt GPT-4 and generate a single CF for each of the six possible interventions (3 writer's concepts with two possible values that are different from the assignment). The prompt is provided in Figure 11. In addition, we ask GPT-4 to predict the new label of the tweet (after the edit) and add this information to the dataset.

Table 6: Results for the new stance detection setup we introduce in an out-of-distribution scenario (for each test example, the candidate set compromise texts with different subjects). We explain two stance detection models: **New** - RoBERTa fine-tuned on new stance labels extracted by GPT-4; and **Original** - DistilBERT fine-tuned with the stance labels of the original tweets. The description of the rows is given in the caption of Table 2. **Numbers are the mean over 18 interventions, ↓ is better.**

| | $K = 1$ | | | | | | $K = 10$ | | | | | | |
| | New Labels | | | Original | | | New Labels | | | Original | | | AVG |
| | *L2* | *Cos* | *ND* | *L2* | *Cos* | *ND* | *L2* | *Cos* | *ND* | *L2* | *Cos* | *ND* | |
|---|---|---|---|---|---|---|---|---|---|---|---|---|---|
| Causal Model (+GT) | **.08** | **.16** | **.06** | **.10** | **.13** | **.08** | - | - | - | - | - | - | **0.10** |
| Zero-shot Generative | .22 | .74 | .16 | .36 | .65 | .30 | .19 | .68 | .15 | .34 | .61 | .30 | .39 |
| Random Match | .57 | .85 | .45 | .85 | .77 | .71 | .46 | .82 | .36 | .76 | .73 | .65 | .66 |
| Propensity | .55 | .87 | .44 | .83 | .80 | .70 | .44 | .83 | .34 | .74 | .74 | .64 | .66 |
| Approx | .54 | .87 | .43 | .73 | .75 | **.60** | .43 | .82 | .34 | .65 | .72 | **.56** | .66 |
| PT RoBERTa | .47 | .89 | .36 | .76 | .75 | .63 | .41 | .84 | .31 | .70 | .73 | .60 | .62 |
| PT S-Transformer | .47 | .86 | .36 | .77 | .75 | .64 | **.40** | .81 | **.30** | .73 | .73 | .63 | .62 |
| Causal Model | **.46** | **.84** | **.36** | **.71** | **.74** | **.60** | **.40** | **.80** | **.30** | **.64** | **.71** | .58 | **.59** |

**Experimental details.** We interpret two stance detection models. The first one is a RoBERTa fine-tuned on the new stance labels extracted by GPT-4. The second model is DistilBERT fine-tuned on the stance labels of the original tweets. Noteworthy, the causal effect of the writer's concepts is more pronounced in the second model. This arises because we designed the dataset to have a large correlation between the original labels and the concepts, which the fine-tuned model captures. In contrast, when fine-tuning with the new labels produced by GPT-4, this causal effect becomes weak. This is mainly because GPT-4 tends to predict the neutral label far more frequently than its representation within the original labels (64.8% vs. 25.5%, refer to Table 7), thereby diminishing the correlation we initially established.

In contrast to CEBaB, our new setup enables testing of our causal matching method in an out-of-distribution scenario (Calderon et al., 2023b). A prevalent approach in the stance detection literature is assessing model robustness against distribution shifts arising from changes in the subject of the texts (Hardalov et al., 2021; Deng et al., 2022). For instance, a model might be trained on texts discussing abortion and then tested on texts about climate change. In our context, such a distribution shift might happen when the subject of the test samples is distinct from the subjects within the matching candidate set. Consequently, we conduct experiments to gauge the performance of our method under circumstances where the distribution shift is evident (i.e., for each test example, the matching set compromises only candidates with different subjects).

For the CF generation approach, we employ ChatGPT (and not GPT-4, which generates the ground-truth CFs of the test set) and only with zero-shot prompts. We benchmark our matching method with the same baselines from §4.2 and the same hyperparameters and training procedure described in §G.

**Results.** We report the results on Table 6. As can be seen, we reproduce our main findings from the previous section on the new benchmark, confirming the robustness of our conclusions, even in out-of-distribution settings. Specifically, the generative approach outperforms the matching methods (although consider that in this setup, the ground-truth CFs are also model-generated). In addition, the causal representation model surpasses the other matching baseline and outperforms the generative model when the candidate set contains ground-truth CFs. Finally, the Top-$K$ technique universally improves all the explainers.

## E  FURTHER DISCUSSION

We hope our theoretical and empirical results might broadly impact the field, and suggest these four following directions:

**Explanation methods should be causal-inspired.** Throughout this paper, our discussions as well as the theoretical and empirical findings, collectively underscore a pivotal message: True interpretability is inextricably linked with causality. As NLP systems become part of our daily lives and influence critical decision-making processes in sectors (like law, healthcare, politics, and education), the

imperative for faithful explanations has never been greater. This realization calls for a paradigm shift towards exploring and developing more causal-inspired explanation methods.

**Understanding when LLM-generated counterfactuals fail.** Our empirical results demonstrate that the best method for explaining the causal effect of high-level concepts on model prediction is by employing multiple (Top-$K$) LLM-generated CFs. However, this does not mean that the problem of interpretability has been solved. There is an impending need for the community to design more challenging benchmarks to identify the areas where the LLMs fall short, misunderstand causality, and fail to provide correct CF explanations (for example, see Kiciman et al. (2023)). Moreover, as we emphasize throughout the paper, utilizing an LLM at inference time is frequently impractical because of its prolonged latency, high financial costs, or privacy concerns, which leads us to the next point.

**Closing the gap of efficient explanation methods.** As the pursuit of understanding model behavior intensifies, it is paramount that we do not trade off efficiency for quality. The noticeable performance gap between the two approaches we introduced in the paper (the generative and the matching approaches) underscores a ripe avenue for research. For example, the candidate set is crucial for the success of the matching approach, and given the right examples within this set, it has the potential to surpass its generative counterpart. Therefore, an interesting line of work could explore techniques for enriching the candidate set. Bridging this gap could lead to methods that encapsulate the best of both worlds: high quality and operational efficiency.

**Constructing new CF-based benchmarks.** As we shift to causal-inspired explanation methods, how we evaluate them must evolve in tandem. Accordingly, it is imperative to benchmark explanations against the true causal effect. High-quality CFs, which can serve as a proxy for ground truth in this context, are the basis for such evaluation, as exemplified by the CEBaB benchmark (Abraham et al., 2022). Nevertheless, the traditional approach for constructing such benchmarks relies on human experts and is both financially costly, labor-intensive and fraught with inherent difficulties due to the cognitive demands of the task. Our work demonstrates that LLMs can effectively facilitate this process. We encourage the community to craft benchmarks that resonate with real-world scenarios. These LLM-guided benchmarks can serve as catalyzers for appraising efficient methods as we work towards bridging the existing performance gap.

# F    REQUIREMENTS AND LIMITATIONS

In this section, we discuss the setup requirements for estimating the causal effect according to our two approaches and the limitations these requirements may pose. In addition, we discuss and demonstrate the applicability of our approaches to more complex causal graphs and setups than CEBaB.

## F.1    REQUIREMENTS

**Causal graph.** The first requirement for any causal estimation method is access to a *causal graph* that describes our causal beliefs, i.e., the concepts and the relationships between them. Notice that we do not assume access to the complete *data-generating process (DGP)*, which besides the causal graph, also describes the exact mathematical model that quantitatively defines the relationships and interactions among concepts and the exogenous variables. Furthermore, the causal effect it must be identifiable from the graph (for an example of a causal graph that is non-identifiable, see Figure 4 in Cinelli et al. (2022)), and therefore, in this study, we discuss only identifiable causal graphs.

To estimate the causal effect given a causal graph, we first need to find the adjustment set (see the back-door criterion in section §H). Complementary, the causal graph also implies which variables should not be adjusted: Mediators (variables that are part of the causal pathway) and Colliders (variables that are caused by both the treatment and the outcome). Adjusting these variables can lead to incorrect estimations. In the case we are interested in the direct causal effect (and not the total effect) of the treatment variable, we can add to the adjustment set the mediators (but not colliders! e.g., *Sore Throat* in Figure F.2).

This fundamental causal concept of adjustment is modeled in our approaches in the following ways. In the generative approach, we specify the adjusted variables in the prompt and optionally also specify non-adjusted variables that may or may not be changed due to the intervention. In the causal

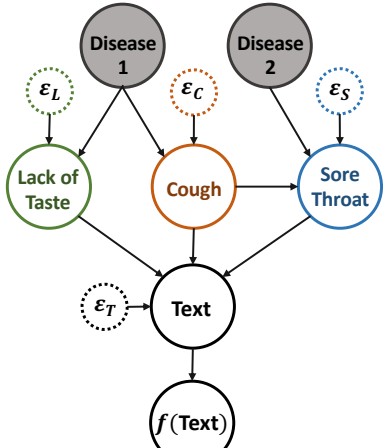

An original health query written by a patient describing its symptoms.

Over the past few days, I've noticed that my cough has become quite severe. It's persistent and seems to be more intense than a usual cough. Along with this, I've started experiencing a sore throat. It's uncomfortable and makes swallowing a bit painful. Another strange thing I've noticed is a change in my sense of taste. Foods don't taste the same as they used to; it's like the flavors are muted or just a little off.

Figure 5: The causal graph of our motivating case study of a patient-doctor interaction in an online health consultation. A text-based health query ($Text$, see example on the right) is analyzed by a doctor with the assistance of an NLP classifier that outputs a diagnosis of possible diseases. In this causal graph, three symptoms affect the patient's text, with a notable interrelationship between $Cough$ and $SoreThroat$. The exogenous variables impacting the symptoms are $\varepsilon_L, \varepsilon_C, \varepsilon_S, \varepsilon_T$.

representation learning method, the adjusted variables are used for constructing the $\mathbb{X}_M$ and $\mathbb{X}_{\neg M}$ sets, which are part of the optimization objective (see Eq.5).

Since the causal graph is typically derived by a domain expert, specifying it might seem like a limitation preventing generalization and automation of our explanation methods. However, notice that causal graph specification is a requirement for accurate causal estimation (Shpitser & Pearl, 2006; Feder et al., 2021a; 2022), this also aligns with our theoretical result: Explanation methods that are unaware of the causal mechanism might be unfaithful. In some fields (such as healthcare), causal explanations are crucial, making the effort to construct a causal graph a worthwhile endeavour. Consider an NLP model that recommends medical treatments based on symptoms described in the medical record. In that case, the clinician can not rely on correlational interpretations of the model recommendations. Finally, following future research and technological advances, it is reasonable to assume that the reliance on human experts for concept and causal graph discovery will reduce, and this process will gradually become more and more automatic.

**Annotated training dataset** In the generative approach, a training dataset is not essential, except when performing few-shot prompting. In this case, a small selection of text examples, interventions, and counterfactuals is required.

Conversely, the general requirement of any matching method is the availability of a candidate set from which the method selects a match. The candidate set should contain an annotation of the treatment concept. This is because the method needs to select a match annotated with the corresponding target intervention value; otherwise, it will fail to provide an accurate estimation. In our setup, the candidate set is randomly sampled from the training set.

For training our causal representation method, we also need a training set annotated with the adjusted variables. This set is used for: (1) Constructing the $\mathbb{X}_{CF}$ and $\mathbb{X}_{\neg CF}$ sets by prompting an LLM to generate CFs of examples from the training set; (2) Training concept predictors - which are used for filtering misspecified CFs; (3) Constructing the $\mathbb{X}_M$ and $\mathbb{X}_{\neg M}$ sets.

Although a candidate set and a training dataset with annotations are used in our experiments, we believe they are not required and can be easily generated by an LLM. We demonstrate this approach in our ablation study (Appendix §C), where we train our matching method with concept annotations predicted by an LLM. According to the results, the performance remains consistent and on par with models trained on human annotations.

## F.2 COMPLEX CAUSAL GRAPHS

Our choice of CEBaB was driven by its unique status as the only non-synthetic interventional dataset tailored for benchmarking concept-level explanation methods in NLP. In this subsection, we discuss and elaborate on the applicability of our theorem and methods to other causal graphs beyond CEBaB.

First, our theorem is designed to be flexible and does not rely on a specific type of causal graph. Our proof is based on a minor modification to the graph, adding a confounder, which can be applied to graphs of any complexity. The key observation is that while non-causal methods may struggle with this change, causal methods remain faithful. Thus, our theorem holds regardless of the complexity of the causal graph as long the causal effect can be identified from the graph.

Second, our methods are also applicable to any identifiable causal graph. The identification requirement is the only limitation of applying our methods. However, without further assumptions on the causal mechanism, it is also a limitation of any causal estimation method. For an example of a causal graph that is non-identifiable see Figure 4 in Cinelli et al. (2022). Given a causal graph, we can find the adjustment set (see the back door criterion in subsection §H), the mediators - descendants which are not colliders and lie in the causal paths between the treatment and the text, and the colliders. These sets are the only deliverable of the causal graph that impact the modeling of our methods (both the generative and the causal matching approaches), and we describe how to use these sets in subsections §3.3 and §F.1.

We next provide proof of concepts that strong LLMs (such as GPT-4) can succeed in generating CFs for graphs that differ from CEBaB, especially those involving mediators. By incorporating a mediator into the causal graph, we aim to explore situations where an intervention leads to changes in more than just the outcome text. Our case study is based on the causal graph illustrated in Figure 5 and involves an online health consultation between a patient and a doctor. The patient submits a written health query, and the doctor then reviews this query, aided by an NLP classifier that estimates the likelihood of various diseases. Based on an explanation method that quantifies the effect of each symptom on the prediction of the disease, the doctor can continue her interaction with the patient and concentrate on specific symptoms for a more accurate diagnosis.

According to the causal graph in Figure 5, there are three symptoms: *Lack of Taste*, *Cough*, and *Sore Throat*. While *Disease 1* causes a Lack of Taste and Cough, *Disease 2* causes a Sore Throat. However, any combination of disease and symptoms is possible due to the involvement of noise originating from the exogenous variables. In addition, an intensive cough can also cause a sore throat. Hence, when Cough is the concept we explain, Lack of Taste is a confounder, while Sore Throat is a mediator. The original query of the patient is provided on the right side of Figure 5, describing a patient suffering from all three symptoms.

In the examples below, we show how we can utilize GPT-4 to generate CFs by intervening on the cough symptom to estimate the direct causal effect (red box) or the total causal effect (green box).

> **GPT-4 prompt for direct effect estimation:** Generate a counterfactual by making the patient's cough weaker. Keep the symptom of lack of taste and sore throat fixed.
>
> **Ex1:** Over the past few days, I've noticed **a mild cough. It's not very persistent and seems less intense than a usual cough.** Along with this, I've started experiencing a sore throat. It's uncomfortable and makes swallowing a bit painful. Another strange thing I've noticed is a change in my sense of taste. Foods don't taste the same as they used to; it's like the flavors are muted or just a little off.
>
> - - - - - - - - - - - - - - - - - - - - - - - - - - - - - - - - - - - - - - - - - - - - - -
>
> **Ex2:** Over the past few days, I've noticed **a mild cough. It's intermittent and not as intense as a usual cough.** Along with this, I've started experiencing a sore throat. It's uncomfortable and makes swallowing a bit painful. Another strange thing I've noticed is a change in my sense of taste. Foods don't taste the same as they used to; it's like the flavors are muted or just a little off.

> **GPT-4 prompt for total effect estimation:** Generate a counterfactual by making the patient's cough weaker. Keep the symptom of lack of taste fixed. Notice that a cough could impact the sore throat.
>
> **Ex1:** Over the past few days, I've noticed **a mild cough. It's not very persistent and seems less intense than a usual cough. Despite the mildness of the cough, I've started experiencing a sore throat.** It's uncomfortable and makes swallowing a bit painful, though not severely so. What's more peculiar is a change in my sense of taste. Foods don't taste the same as they used to; it's like the flavors are muted or just a little off.
>
> - - - - - - - - - - - - - - - - - - - - - - - - - - - - - - - - - - - - - - - - - - - -
>
> **Ex2:** Over the past few days, I've noticed that **my cough is present but not particularly severe. It's more like a mild, occasional cough rather than a persistent or intense one. Along with this, there's a slight soreness in my throat, but it's not too uncomfortable and doesn't cause much pain when swallowing.** Another strange thing I've noticed is a change in my sense of taste. Foods don't taste the same as they used to; it's like the flavors are muted or just a little off.

To generate approximate CFs for estimating the direct causal effect (red box), we adjust for the Lack of Taste and the Sore Throat concepts since we are only interested in the direct path between Cough and the text. We do this by prompting GPT-4 to keep these variables fixed. The red box shows GPT-4 successfully generates CFs by modifying the text describing the cough symptom while leaving the remaining text fixed.

In the case of total effect estimation (green box), we should not adjust for the mediator concept - Sore Throat. This is because a change in Cough may or may not cause a change in Sore Throat (which also affects the text). We can achieve this by informing GPT-4 that a change in cough could also impact sore throat. As the two CFs in the green box show, GPT-4 sometimes ignores a potential change in the sore throat symptom (Ex1) and sometimes modifies the relevant text (Ex2). This demonstrates that strong LLMs like GPT-4 can handle complex situations when an intervention not only directly affects the text but also may impact other concepts.

Notice, however, that relying on a single CF might lead to a biased causal effect estimation. Therefore, generating multiple CFs is crucial. This perfectly aligns with our theoretical definition of an approximated CF explanation method which considers an approximation error. Accordingly, utilizing multiple approximated CF for estimating the causal effect may lower the variance of the estimator and make it more robust. This also explains why this technique (of Top-K matching) universally improves the performance of any CF-based explanation method in our study. Notice that in the case of mediators, LLMs may not model accurately the conditional distribution of the mediator given the intervention (e.g., $P(SoreThroat|Cough)$ in our causal graph). We leave this challenge to future research, although a solution might be reweighing the CFs according to this probability.

Finally, our work underscores a novel contribution in utilizing LLMs for generating CFs. As we demonstrate, this can facilitate the creation of new interventional datasets representing complex causal graphs and significantly advance the research in causal explanations and benchmark construction.

## G  IMPLEMENTATION DETAILS

### G.1  CAUSAL EFFECT ESTIMATION PIPELINE

This study focuses on black-box NLP model explanations by estimating the causal effect of high-level concepts on model prediction. The aim of this subsection is to provide additional details about the causal estimation pipeline performed according to the two approaches for counterfactual approximation proposed in Section §3: The generative approach and the matching approach.

Given a query (treated) example $\boldsymbol{x}_t$, an intervention $T : t \to t'$ and an approximated CF $\widetilde{\boldsymbol{x}}_{t'}^m$, we use the following estimator for estimating the individual concept effect on a classifier $f$:

$$\widehat{\text{ICaCE}}_f(\boldsymbol{x}_t, T, t, t') := \widehat{\text{ICaCE}}_f(\boldsymbol{x}_t, \widetilde{\boldsymbol{x}}_{t'}^m) = f(\widetilde{\boldsymbol{x}}_{t'}^m) - f(\boldsymbol{x}_t)$$

In addition, we can provide a more robust estimator (see the third findings in our results §5) by performing *Top-K matching*:

$$\widehat{\texttt{ICaCE}}_f(\boldsymbol{x}_t, T, t, t') = \frac{1}{K}\sum_{i=1}^{K}\widehat{\texttt{ICaCE}}_f(\boldsymbol{x}_t, \widetilde{\boldsymbol{x}}_{t'}^{i})$$

After calculating the $\widehat{\texttt{ICaCE}}_f$, we can also estimate the $\widehat{\texttt{CaCE}}_f$ using:

$$\widehat{\texttt{CaCE}}_f(T, t, t') = \frac{1}{|\mathbb{D}|}\left(\sum_{\boldsymbol{x}_t}\widehat{\texttt{ICaCE}}_f(\boldsymbol{x}_t, \widetilde{\boldsymbol{x}}_{t'}^{m}) + \sum_{\boldsymbol{x}_{t'}}\widehat{\texttt{ICaCE}}_f(\widetilde{\boldsymbol{x}}_t^{m}, \boldsymbol{x}_{t'}) + \sum_{\boldsymbol{x}}\widehat{\texttt{ICaCE}}_f(\widetilde{\boldsymbol{x}}_t^{m}, \widetilde{\boldsymbol{x}}_{t'}^{m})\right)$$

We next describe the two approaches for counterfactual approximation.

**The generative approach.** Given a query example $\boldsymbol{x}_t$, and an intervention $T : t \to t'$, we prompt the LLM to generate a CF. The prompt instructs the LLM to intervene on the text by changing the value of the concept $T$ from $t$ to $t'$. For the Top-K technique we simply generate multiple CFs.

**The matching approach.** Given a query example $\boldsymbol{x}_t$, and an intervention $T : t \to t'$, the matching approach aim to select a match from the candidate set $\mathbb{D}(T = t')$, which contains a subset of examples such that their treated concept $T$ is equal to the target intervention value $t'$.

In this study, the selection of a match is performed by computing the similarity between the representations of the query example and each of the candidate set. An exception are the Random matching and the Approx baselines. Accordingly, the matching method rank the candidates according to their similarity and selects the most similar one:

$$\widetilde{\boldsymbol{x}}_{t'}^{m} = m_1(\boldsymbol{x}_t) = \underset{\boldsymbol{x}_{t'} \in \mathbb{D}(T=t')}{\arg\max} s\left(\phi(\boldsymbol{x}_t), \phi(\boldsymbol{x}_{t'})\right) \tag{4}$$

The high-dimensional representation $\phi(\cdot)$ of each text is extracted by a text encoder and is the mean vector of the last layer hidden states of the text's tokens. The similarity function $s(\cdot)$ used in our study is the cosine similarity:

$$s(\phi(\boldsymbol{x}_t), \phi(\boldsymbol{x}_{t'})) = \frac{\phi(\boldsymbol{x}_t)^T\phi(\boldsymbol{x}_{t'})}{\|\phi(\boldsymbol{x}_t)\| \cdot \|\phi(\boldsymbol{x}_{t'})\|} \tag{11}$$

The matching approach proposed in this paper utilizes a causal representation model that is trained using the procedure described below (Appendix §G.2).

### G.2 TRAINING PROCEDURE

We start with a small set of textual examples (the train set) labeled with the adjusting variables (e.g., review concepts such as food and service). Initially, we employ a small Encoder-only model (RoBERTa) and train a concept predictor for each variable. The predictor models are then harnessed to construct the four sets: By predicting the concept values of each training example, we divide them (according to their concept values) and construct the $\mathbb{X}_{\text{M}}$ and $\mathbb{X}_{\neg\text{M}}$ sets.

For constructing $\mathbb{X}_{\text{CF}}$ and $\mathbb{X}_{\neg\text{CF}}$, we use a few-shot LLM (ChatGPT) to generate approximate CFs and misspecified CFs (see prompts at §I). We filter out misspecified CFs from $\mathbb{X}_{\text{CF}}$ if the concept predictors indicate an adjusted variable was also changed during the treatment intervention. Additionally, we use simple rules to filter unsuccessful generations (e.g., empty strings, "As an AI LM...").

To train our language representation model (aimed to explain a specific concept), we proceed as follows: For every training example $\boldsymbol{x}_t$, we randomly sample four examples: $\boldsymbol{x}_{\text{CF}} \in \mathbb{X}_{\text{CF}}, \boldsymbol{x}_{\text{M}} \in \mathbb{X}_{\text{M}}, \boldsymbol{x}_{\neg\text{CF}} \in \mathbb{X}_{\neg\text{CF}}$. The training goal is to minimize the objective of Eq. 5. We repeat this process for several epochs (15) and select the checkpoint that achieves the lowest loss on the development set.

### G.3 HYPERPARAMETERS

We use $\tau = 0.1$, train the causal representation models for 12 epochs, and select the model checkpoint that minimizes the objective of Eq. 5 (or its modified version for the ablation models). Notice that

when using the Err as the selection criterion, one might be able to select a more robust checkpoint. However, this violates the model-agnostic property, as calculating the Err requires access to the explained model.

The backbone SentenceTransformer of our model is MPNet (all-mpnet-base-v2) (Song et al., 2020). We use a learning rate of 5e-6 for training the causal representation models. We use a learning rate 1e-5 and a batch size 16 for the fine-tuned baselines, explained models and concept predictors. The concept predictor models are used for filtering misspecified CFs (for constructing the $\mathbb{X}_{\text{CF}}$ and $\mathbb{X}_{\neg\text{CF}}$ sets) and for the Approx and Propensity score baselines. We fine-tune a RoBERTa model using the annotated train set for each of the four concepts.

For generating counterfactuals, we use the OpenAI API with the following models: gpt-3.5-turbo (referred to as ChatGPT in the paper) and gpt-4 (GPT-4).

### G.4 EXPLANATION METHOD EVALUATION PIPELINE

A high-level explanation method that provides an estimator of the ICaCE can be evaluated using benchmarks like CEBaB, i.e., an interventional dataset consisting of examples, interventions, and a corresponding human-written ground-truth CF for each example and intervention. For a given example $\boldsymbol{x}_t$ and an intervention $T : t \to t'$, using the example and its corresponding ground-truth CF $\widetilde{\boldsymbol{x}}_{t'}^h$, we can estimate the golden individual causal effect:

$$y^h = \widehat{\text{ICaCE}}_f(\boldsymbol{x}_t, \widetilde{\boldsymbol{x}}_{t'}^h) = f(\widetilde{\boldsymbol{x}}_{t'}^h) - f(\boldsymbol{x}_t)$$

For the same example $\boldsymbol{x}_t$ and intervention, the explanation method estimates the individual causal effect $y^m$. In the case of a CF-based explanation method, the estimation is computed with the approximated CF: $y^m = \widehat{\text{ICaCE}}_f(\boldsymbol{x}_t, \widetilde{\boldsymbol{x}}_{t'}^m)$. In this study, the approximated CF is an LLM-generated CF or a match.

Given the two estimations, $y^h$ and $y^m$, we then calculate the distance between the two using three metrics introduced in Abraham et al. (2022) $L2$ distance, Cosine distance, and norm difference.

$$\text{L2}(y^h, y^m) = \|y^h - y^m\|_2 \tag{12}$$

$$\text{Cos}(y^h, y^m) = 1 - \frac{(y^h)^T y^m}{\|y^h\|_2 \|y^m\|_2} \tag{13}$$

$$\text{ND}(y^h, y^m) = \left| \|y^h\|_2 - \|y^m\|_2 \right| \tag{14}$$

Finally, the distance is plugged into Eq.7 to provide the estimation error of the explanation method:

$$\text{Err}(f, m, T, t, t') = \frac{1}{|\mathbb{D}(T \leftarrow t')|} \sum_{(\boldsymbol{x}_t, \widetilde{\boldsymbol{x}}_{t'}^h) \in \mathbb{D}} \text{Dist}\left( \widehat{\text{ICaCE}}_f(\boldsymbol{x}_t, \widetilde{\boldsymbol{x}}_{t'}^h), \widehat{\text{ICaCE}}_f(\boldsymbol{x}_t, \widetilde{\boldsymbol{x}}_{t'}^m) \right) \tag{7}$$

## H  FINDING THE ADJUSTMENT SET

**Causal paths.** The following three triplets (or $X$, $Y$ and $Z$) are the main patterns of a causal graph:

- Chains (or mediators): $X \to Z \to Y$. Conditioning on $Z$ blocks the flow from $X$ to $Y$.

- Forks (or common causes, confounders): $X \leftarrow Z \to Y$. Conditioning on $Z$ blocks the flow from $X$ to $Y$.

- Colliders (or common effects): $X \to Z \leftarrow Y$. The flow from $X$ to $Y$ is blocked by default. However, conditioning on $Z$ opens the flow and induces an association between $X$ and $Y$.

**Back-door criterion (Pearl, 2009).** A set of variables $\mathbb{Z}$ satisfies the back-door criterion relative to $(X, Y)$ in a directed acyclic graph $\mathcal{G}$ if:

- No node in $\mathbb{Z}$ is a descendant of $X$.

- $Z$ blocks every path between $X$ and $Y$ that contains an arrow into $X$.

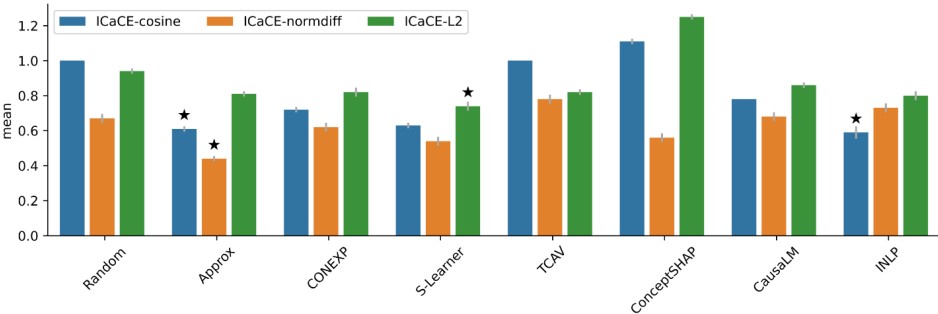

Figure 6: A comparison between different explanation methods adapted from Abraham et al. (2022). The best performing methods are the simple Approx baseline and the S-learner.

The adjustment criterion (Shpitser et al., 2010) was later devised to handle cases in which Z may explicitly contain descendants of $X$; however, it is unnecessary for our causal graph described in Figure 1. We term the set $\mathbb{Z}$, which satisfies the back-door criterion as the *adjustment set*.

**The adjustment set of CEBaB:** In our causal graph (Figure 1), when estimating the causal effect of an aspect, for example, $S$, on the model $f$, the following paths should be taken into account ($F$ and $N$ w.l.o.g):

- $S \rightarrow X \rightarrow f(X)$
- $S \leftarrow U \rightarrow F \rightarrow X \rightarrow f(X)$
- $S \leftarrow U \rightarrow F \rightarrow Y \leftarrow N \rightarrow X \rightarrow f(X)$
- $S \rightarrow Y \leftarrow F \rightarrow X \rightarrow f(X)$
- $S \rightarrow Y \leftarrow F \leftarrow U \rightarrow N \rightarrow X \rightarrow f(X)$
- (when $Y \rightarrow X$) $S \rightarrow Y \rightarrow X \rightarrow f(X)$
- (when $Y \leftarrow X$) $S \rightarrow Y \leftarrow X \rightarrow f(X)$

The adjustment set is $F$, $N$, $A$, and $Y$ must not be adjusted. Accordingly, the set of matches $\mathbb{X}_{\mathsf{M}}$ should contain texts with the same aspect values (excluding the treatment) as the query (treated) example. Moreover, it clarifies why we use the term misspecified to describe the sets $\mathbb{X}_{\neg\mathsf{M}}$ and $\mathbb{X}_{\neg\mathsf{CF}}$. This is because they contain texts that at least one of their aspect values is different from the query example or was changed to a different one.

**Increased precision when conditioning on $V$.** The exogenous variable $V$ is a direct cause of the text $X$ that mediates between concepts (e.g., $S$) and the model prediction $f(X)$. For example, the variable $V$ can account for the syntax, writing style, or length of $X$. Controlling for $V$ (hypothetically, since it is not observed) will not bias the causal effect estimation (of CaCE, and not of ICaCE, which by definition should control for $V$) since it does not open any back-door paths from $S$ to $Y$. On the other hand, controlling for $V$ can increase the estimation precision of CaCE (Cinelli et al., 2022).

Consider the case where the model $f$ learns spurious correlations between $V$ and $Y$ (e.g., the sentiment of long texts tends to be negative). We use the Approx matching technique (which controls all the adjusted variables: $F$, $N$, and $A$ and sample a unit from the control group which shares the same adjustment values as the query example) to calculate the $\widehat{\mathsf{CaCE}}_f$ (see Eq.2). Since $V$ is independent of any other variable (except $X$), the $\widehat{\mathsf{CaCE}}_f$ estimator is not biased asymptotically. However, when the candidate sets are finite (and small), the non-representing sample might amplify the spurious correlations $f$ learns. Controlling for $V$ (finding matches that also share the same attributes, e.g., writing style) mitigates this bias and increases the precision of the estimation.

Therefore, utilizing counterfactuals while training the causal representation model used for matching is vital. In contrast to $\mathbb{X}_{\mathsf{M}}$, the set of counterfactuals $\mathbb{X}_{\mathsf{CF}}$ should share the same values of $V$, and the causal model learns to consider it when finding a match.

## I   PROMPTS AND EXAMPLES

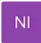

Rate the restaurant on a scale from 1 to 5.
review:
[TEXT]

rating:

Figure 7: The prompt we use for rating CEBaB reviews and extracting the five-way sentiment distribution (of the Llamas models). We extract the next token probabilities of 1-5 tokens. [TEXT] is a review.

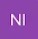

I'm providing a review from OpenTable and your task is to edit the review according to my instruction.
---- Input Review ----
[TEXT]
----- Instruction -----
The reviewer gave a POSITIVE score to the FOOD aspect. You should edit the review such that there is a NEGATIVE score to the FOOD aspect.
We consider four aspects FOOD, AMBIANCE, SERVICE, NOISE, make sure you change ONLY the rating of FOOD.
----- Edited Review -----

Figure 8: The zero-shot prompt we use for generating CFs for CEBaB examples. [TEXT] is the review. The instruction is changed according to the treatment (e.g., in this prompt, we change the value of the food aspect from positive to negative.

Table 7: Fractions in percentages of the variables (joint probability) in the new setup (Stance Detection, §D). For example, according to row **CC** and column **T**, 7.3% of the data are tweets about Climate Change (Subject concept) written by a teenager (Age concept). There are a total of 4000 examples divided into the train ($N = 1000$), matching ($N = 2250$), dev ($N = 250$), and test ($N = 500$) sets. **Orig. Label** is the label of the original tweet before any modifications, and **Label** is the label predicted by GPT-4 after modifying the tweet. Legend: *Subject* – **Ab** (Abortions), **At** (Atheism), **CC** (Climate Change) **Fe** (Feminism) **HC** (Hillary Clinton); *Age* – **?** (Unknown), **T** (Teenager), **E** (Elder); *Job* – **?** (Unknown), ♀ (Female), ♂ (Male); *Job* – **?** (Unknown), **F** (Farmer), **P** (Professor); *Label* – **?** (Neutral), **X** (Oppose), ✓ (Support); *Orig. Label* – **?** (Neutral), **X** (Oppose), ✓ (Support).

| | Subject | | | | | Age | | | Gender | | | Job | | | Label | | | Orig. Label | | |
|---|---|---|---|---|---|---|---|---|---|---|---|---|---|---|---|---|---|---|---|---|
| | Ab | At | CC | Fe | HC | ? | T | E | ? | ♀ | ♂ | ? | F | P | ? | ✗ | ✓ | ? | ✗ | ✓ |
| **Ab** | 20.0 | 0.0 | 0.0 | 0.0 | 0.0 | 7.2 | 3.6 | 9.1 | 10.1 | 3.8 | 6.2 | 8.6 | 7.2 | 4.1 | 13.4 | 5.2 | 1.4 | 4.9 | 11.6 | 3.5 |
| **At** | 0.0 | 20.0 | 0.0 | 0.0 | 0.0 | 7.7 | 5.4 | 7.0 | 9.7 | 4.2 | 6.0 | 9.7 | 6.5 | 3.8 | 13.1 | 5.6 | 1.3 | 4.0 | 12.4 | 3.6 |
| **CC** | 0.0 | 0.0 | 20.0 | 0.0 | 0.0 | 8.7 | 7.3 | 4.0 | 9.5 | 6.2 | 4.3 | 8.4 | 4.0 | 7.6 | 12.5 | 1.0 | 6.5 | 7.6 | 0.8 | 11.5 |
| **Fe** | 0.0 | 0.0 | 0.0 | 20.0 | 0.0 | 9.2 | 5.4 | 5.4 | 8.3 | 4.6 | 7.2 | 8.4 | 6.6 | 4.9 | 13.1 | 1.8 | 5.0 | 3.4 | 10.8 | 5.8 |
| **HC** | 0.0 | 0.0 | 0.0 | 0.0 | 20.0 | 8.2 | 4.5 | 7.3 | 9.4 | 4.1 | 6.5 | 8.0 | 8.4 | 3.6 | 12.7 | 5.3 | 2.0 | 5.5 | 11.2 | 3.3 |
| **?** | 7.2 | 7.7 | 8.7 | 9.2 | 8.2 | 41.0 | 0.0 | 0.0 | 11.2 | 13.0 | 16.8 | 8.4 | 18.1 | 14.6 | 28.2 | 6.5 | 6.3 | 11.0 | 18.9 | 11.1 |
| **T** | 3.6 | 5.4 | 7.3 | 5.4 | 4.5 | 0.0 | 26.2 | 0.0 | 14.9 | 6.2 | 5.2 | 16.6 | 4.2 | 5.4 | 16.5 | 2.6 | 7.1 | 8.7 | 4.9 | 12.6 |
| **E** | 9.1 | 7.0 | 4.0 | 5.4 | 7.3 | 0.0 | 0.0 | 32.8 | 20.9 | 3.6 | 8.2 | 18.2 | 10.4 | 4.1 | 20.1 | 9.9 | 2.8 | 5.8 | 23.0 | 4.0 |
| **?** | 10.1 | 9.7 | 9.5 | 8.3 | 9.4 | 11.2 | 14.9 | 20.9 | 47.0 | 0.0 | 0.0 | 11.6 | 21.2 | 14.1 | 30.4 | 9.8 | 6.8 | 11.4 | 23.3 | 12.3 |
| **♀** | 3.8 | 4.2 | 6.2 | 4.6 | 4.1 | 13.0 | 6.2 | 3.6 | 0.0 | 22.9 | 0.0 | 14.4 | 3.4 | 5.1 | 14.0 | 2.6 | 6.3 | 6.1 | 6.4 | 10.4 |
| **♂** | 6.2 | 6.0 | 4.3 | 7.2 | 6.5 | 16.8 | 5.2 | 8.2 | 0.0 | 0.0 | 30.1 | 17.2 | 8.2 | 4.8 | 20.6 | 6.5 | 3.1 | 8.0 | 17.2 | 5.0 |
| **?** | 8.6 | 9.7 | 8.4 | 8.4 | 8.0 | 8.4 | 16.6 | 18.2 | 11.6 | 14.4 | 17.2 | 43.2 | 0.0 | 0.0 | 26.0 | 9.1 | 8.2 | 11.4 | 19.0 | 12.7 |
| **F** | 7.2 | 6.5 | 4.0 | 6.6 | 8.4 | 18.1 | 4.2 | 10.4 | 21.2 | 3.4 | 8.2 | 0.0 | 32.8 | 0.0 | 22.1 | 8.2 | 2.5 | 8.5 | 21.8 | 2.5 |
| **P** | 4.1 | 3.8 | 7.6 | 4.9 | 3.6 | 14.6 | 5.4 | 4.1 | 14.1 | 5.1 | 4.8 | 0.0 | 0.0 | 24.0 | 16.8 | 1.6 | 5.6 | 5.5 | 6.0 | 12.5 |
| **?** | 13.4 | 13.1 | 12.5 | 13.1 | 12.7 | 28.2 | 16.5 | 20.1 | 30.4 | 14.0 | 20.6 | 26.0 | 22.1 | 16.8 | 64.8 | 0.0 | 0.0 | 24.3 | 27.2 | 13.3 |
| **✗** | 5.2 | 5.6 | 1.0 | 1.8 | 5.3 | 6.5 | 2.6 | 9.9 | 9.8 | 2.6 | 6.5 | 9.1 | 8.2 | 1.6 | 0.0 | 19.0 | 0.0 | 0.5 | 17.4 | 1.1 |
| **✓** | 1.4 | 1.3 | 6.5 | 5.0 | 2.0 | 6.3 | 7.1 | 2.8 | 6.8 | 6.3 | 3.1 | 8.2 | 2.5 | 5.6 | 0.0 | 0.0 | 16.2 | 0.7 | 2.2 | 13.3 |
| **?** | 4.9 | 4.0 | 7.6 | 3.4 | 5.5 | 11.0 | 8.7 | 5.8 | 11.4 | 6.1 | 8.0 | 11.4 | 8.5 | 5.5 | 24.3 | 0.5 | 0.7 | 25.5 | 0.0 | 0.0 |
| **✗** | 11.6 | 12.4 | 0.8 | 10.8 | 11.2 | 18.9 | 4.9 | 23.0 | 23.3 | 6.4 | 17.2 | 19.0 | 21.8 | 6.0 | 27.2 | 17.4 | 2.2 | 0.0 | 46.8 | 0.0 |
| **✓** | 3.5 | 3.6 | 11.5 | 5.8 | 3.3 | 11.1 | 12.6 | 4.0 | 12.3 | 10.4 | 5.0 | 12.7 | 2.5 | 12.5 | 13.3 | 1.1 | 13.3 | 0.0 | 0.0 | 27.7 |

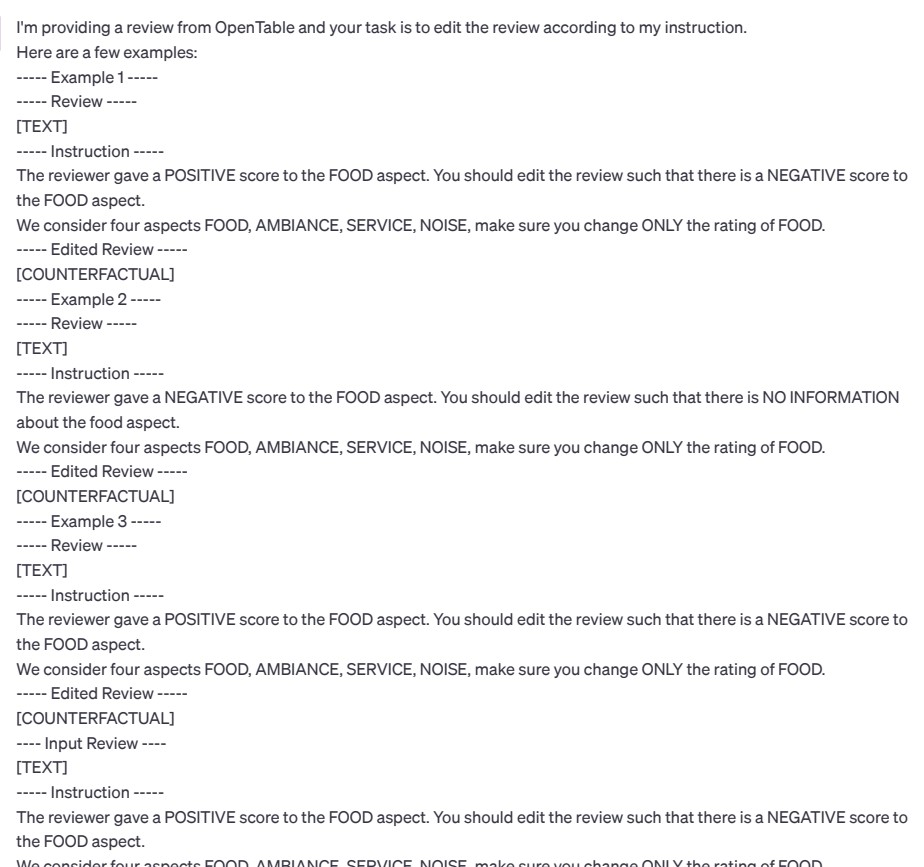

Figure 9: The few-shot prompt we use for generating CFs for CEBaB examples. `[TEXT]` is a review and `[COUNTERFACTUAL]` is a human-written CF. Every prompt consists of three demonstrations where the aspect of the treatment is changed. Since there are four aspects, only 12 human-written CFs are required.

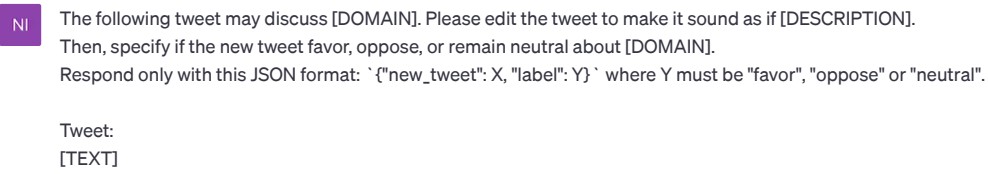

Figure 10: The prompt we use for generating examples for the new stance detection setup. Notice - these examples are used for enriching the dataset with more writer profiles. `[DOMAIN]` is the tweet topic/subject (e.g., abortions), and `[TEXT]` is the original tweet which we ask the LLM (GPT-4) to edit. The `[DESCRIPTION]` describes the new profile of the writer. For example, it can be: "an elder (60+) farmer wrote it". In addition, we ask the LLM to predict the label of the new tweet.

NI  The following tweet may discuss [DOMAIN]. The writer's profile is - [DESCRIPTION].
    Please edit the tweet to make it sound as if [EDIT_DESCRIPTION].
    Then, specify if the new tweet favor, oppose, or remain neutral about [DOMAIN].
    Respond only with this JSON format: `{"new_tweet": X, "label": Y}` where Y must be "favor", "oppose" or "neutral".

    Tweet:
    [TEXT]

Figure 11: The prompt we use for generating ground-truth CFs for examples from the new stance detection setup. `[DOMAIN]` is the tweet topic/subject (e.g., abortions), and `[TEXT]` is the original tweet which we ask the LLM (GPT-4) to edit. The `[DESCRIPTION]` describes the old profile of the writer (e.g., "Age: Unknown, Gender: Female, Job: professor"). The `[EDIT DESCRIPTION]` describes the new profile, for example, "an elder (60+) farmer wrote it", or "the age, gender, or job of the writer is unknown". In addition, we ask the LLM to predict the label of the new tweet.

NI  The following tweet may discuss [DOMAIN]. The writer's profile is - [DESCRIPTION].
    Please edit the tweet to make it sound as if [EDIT_DESCRIPTION].
    Respond only with the revised text.

    Tweet:
    [TEXT]

Figure 12: The zero-shot prompt we use for generating CFs for examples from the new stance detection setup. This prompt is used for benchmarking the LLM (ChatGPT), and not for generating instances for the test set. See Figure 11 caption for more details.

