# OpenReview forum: "Faithful Explanations of Black-box NLP Models Using LLM-generated Counterfactuals"
_ICLR.cc/2024/Conference — ICLR 2024 poster_

### Official Review · Reviewer_GEWc · 2023-10-17

**Soundness:** 3 good
**Presentation:** 2 fair
**Contribution:** 3 good
**Rating:** 6
**Confidence:** 4

**Summary:**

This paper focuses on improving causal explanations, specifically the counterfactual explanation. The authors propose to leverage LLMs to generate the counterfactual input corpus, use the generated corpus to train a counterfactual representation model, and match the input and its corresponding counterfactual representation to generate the causal explanations. Experiment results show that the authors' method outperforms all previous matching baselines, representing a promising explanation ability.

**Strengths:**

1. This paper provides detailed proofs and descriptions of the proposed method.
2. The experiment results are reliable with the comparison between various baselines and models.
3. The authors also construct a benchmark based on the findings of LLM's ability to generate counterfactual examples, which I think is a good contribution to the XAI community.

**Weaknesses:**

The description of the proposed method in Section 3 is confusing and not easy to understand. I think the authors should rephrase Section 3 with a general description of the proposed causal model.

About Eq.(2), the authors use the difference in the model's prediction before and after the treatment as the treatment effect, which, in my opinion, is not robust when the model's output confidence is flat (or the uncertainty is high). This will affect the method's performance on small models like BERT.

**Questions:**

1. I cannot find an accurate definition of "causal model". Does the author use the representation generated by a language representation model optimized with Eq.(5) with generated counterfactual and matched examples, then calculate the matching value in Eq.(4), and use this value to calculate Eq.(2) as a causal model?

2. How does the generative approach work? If (1) is true, does the causal model include the generative approach?

3. There are two versions of ChatGPT (GPT-3.5 and GPT-4); which one did the authors use in the experiments?

---

> ### Author Response · Authors · 2023-11-15
>
> We thank the reviewer for the feedback and advice.
> As recommended by reviewer GEWc, we have introduced two subsections to the appendix including a detailed description of the causal estimation pipeline (G.1) and the evaluation pipeline of explanation methods (G.2).
>
> **Regarding Eq.2:**
>
> This is a smart take. Eq.2 defines an estimator for the ICaCE, which is the causal effect of a concept on the classifier prediction. The estimator is the difference between predictions of an example and its approximated counterfactual (before and after the treatment). Notice that if the expectation of the approximation error is zero, then the estimator is unbiased and measures accurately the ICaCE (see our proof in Appendix A).
>
> We agree that if the outputs of the classifier are flat (or have a high uncertainty), then the difference is small (or has a small norm in case it is a vector). However, so is the real ICaCE - it is not a limitation or weakness of the estimator, but the natural behaviour of the causal effect (see Goyal et.al., 2020)
> For example, when the classifier is random and assigns the same values for every example, then the causal effect would be 0 (i.e., the concept has no effect on the prediction), the same as the estimator. Nevertheless, we agree that when the uncertainty of the model is high, the causal effect is low, and the estimator and the results might be noisy. For this reason, we conducted multiple experiments and explained various models of different sizes and architectures.
>
>
> **Regarding your questions:**
>
> 1. Your description of the estimation of the causal effect is accurate. We use the term “causal model” as a notation for “the causal representation model optimizing Eq.6”. Given an example, we first extract the representations of this example using the causal model. We then also extract the representations of the candidate set (this is done once at training since the candidate set does not change). We then find a match using Eq.4. After finding the match, we estimate the causal effect using Eq.2.
> Aspiring by your questions, we will include a page in the appendix and thoroughly detail the matching, estimation, and evaluation pipelines.
>
> 2. According to the generative approach, we prompt an LLM to generate a counterfactual, and then use Eq.2 to measure the causal effect.
> When we write “causal model” we mean the causal representation model, which is used for matching (see the first question).
> Notice that we do use an LLM to generate counterfactuals for training the causal model. However, we do not generate any counterfactual at inference time when finding a match.
>
> 3. For CEBaB: We use GPT-3.5-turbo for generating all the counterfactuals, for both the generative approach (zero-shot and few-shot, Table 2) and for generating CFs for training the causal model (matching approach).
> For the new Stance Detection dataset: We use GPT-4 for constructing the new dataset (see section 5.1 and Appendix D), but as with CEBaB, we use GPT-3.5-turbo for the generative approach and for training the causal model that we benchmark with the new dataset. We will clarify this in the final version.
>
>
> Dear reviewer, we would like to discuss other weaknesses that you find in our paper with you. We hope we can convince you that this work is important. Please see our summarized contributions:
>
> 1. While the idea of causal explanations being integral to model faithfulness has been discussed in prior works, our paper is the first to provide a theoretical framework that unifies the concepts of faithfulness and causal explanation. Specifically, we prove that CFs (which are a fundamental concept in causal inference) guarantee faithfulness, unlike other model explanation techniques, such as probing, concept removal, adversarial approaches, etc.
>
> 2. In line with recent advancements in the capabilities of LLMs, we are the first to show that LLM-generated CFs provide the SOTA NLP model explanation.
>
> 3. Moreover, we demonstrate empirically that utilizing multiple CFs enhances the quality of the causal estimator across various explanation methodologies. This empirical validation is the first of its kind and is both important and useful.
>
> 4. This is the first paper that thoroughly explores matching techniques for causal model explanation in NLP. Specifically, we enhance the matching paradigm originating in the causal literature such that it considers the rich representation of the text, and not only the observed concepts. This key contribution lays the groundwork for future research in causal representation learning from text.
>
> 5. Finally, recognizing the gap in benchmarks for causal model explanations, we construct a new interventional benchmark. This demonstration paves the way for future research to explore causal model explanations for a broader range of NLP problems and more complex setups.
>
>
> **References:**
>
> Goyal, Yash, et al. Explaining classifiers with causal concept effect (cace). arXiv preprint arXiv:1907.07165, 2019.

---

> > ### Comment · Reviewer_GEWc · 2023-11-21
> > **Response to Authors**
> >
> > Thank you for providing further details of the proposed method. I've carefully checked your rebuttal and have a better understanding of the proposed causal model. Therefore, I will keep my score unchanged and raise my confidence to 4.

---

### Official Review · Reviewer_wvZe · 2023-11-01

**Soundness:** 2 fair
**Presentation:** 2 fair
**Contribution:** 2 fair
**Rating:** 6
**Confidence:** 4

**Summary:**

This paper presents two methods for explaining the predictions of Natural Language Processing (NLP) models, focusing on the use of counterfactual approximations (CFs). The first is a Counterfactual Generation approach, where a large language model (LLM) is prompted to change a specific text concept while keeping others the same. The second is a Matching method that identifies text with similar properties within a dataset. The authors establish the value of approximating CFs for offering _faithful_ explanations and illustrate their techniques' applicability on several models. They further improve the ability to provide explanations using top-K matching. Furthermore, they highlight the potential of LLMs to create new benchmarks for NLP model explanations. The authors present theoretical and empirical evidence to support their research and propose further areas of study.

**Strengths:**

- This paper contributes to the field of NLP model interpretability by introducing two practical methods for model-agnostic explanations, which could improve our understanding of model predictions.
- The authors back their theoretical constructs with extensive experimental results, though the reliability of these methods depends on the specific conditions in which they are applied (such as access to a candidate set that offers good matching candidates).
- The concept of Order-faithfulness is an innovative criterion for explanation methods, potentially providing valuable insights into the relative impact of different concepts on model predictions, although it would need to be tested across various contexts and model types to ensure its broad applicability.

**Weaknesses:**

- The first method proposed, Counterfactual Generation, is computationally expensive and may be infeasible in scenarios requiring real-time explanations.
- Although Matching is faster than CF Generation, it might not be as accurate for all situations, especially when the matching candidate set does not sufficiently represent the input data. It would be great if the authors performed some ablation studies (reducing the quality of the matches in the candidate sets to show how much the performance degrades).
- It's unclear how these techniques would perform on models trained on very niche tasks, which could inherently limit the possible counterfactuals, especially where such attributes may be hard to define beforehand. Both counterfactual generation and matching approaches assume that we have a set of attributes for which we wish to examine whether a model is paying attention to those. However, how does this approach work for open ended tasks (which is where LLMs are primarily being used) or tasks with a large number of classes, where generating counterfactuals (or finding matches) may be inherently difficult?
- It's unclear how these methods would handle situations with multilayered complexities, such as nested counterfactuals, where counterfactual changes to one concept might trigger changes to other related concepts. The paper also does not extensively address scenarios where counterfactual approximations could result in impossibilities or logical contradictions, potentially limiting the breadth of their application.

**Questions:**

I would appreciate if the authors discuss the issues I brought up in the previous section.

Additionally, could you explain the difference between Random Match and Approx again? It's not that clear from the paper.

---

> ### Author Response · Authors · 2023-11-15
>
> Thank you for your valuable feedback. We appreciate your openness to further discussions.
>
> **Regarding the cost of counterfactual generation:**
>
> We agree with this comment and we actually discuss this issue in section 3 and Appendix B. Regardless, this approach may be used in high-stakes applications where accuracy is crucial. For other cases, we develop our efficient matching method, which is much faster and computationally cheaper. We would also like to refer the reviewer to the fact that the costs of strong LLMs like GPT-4 are becoming constantly cheaper (e.g. the recent announcement by OpenAI, which cut these costs by a half).
>
> **Regarding ablations and the quality of the candidate set:**
>
> The performance of matching methods indeed depends on the quality of the candidate set. Notice that we conducted extensive ablation experiments to explore the quality of the matching method as a function of the candidate set quality. While these experiments are in the appendix, they are referred to from the main paper.
>
> 1. In the ablation study (Appendix C, referred to from pages 5 and 7), we evaluated different models across three types of candidate sets: (a) the standard candidate set (randomly sampled from the training set), (b) the standard set augmented by ground-truth CFs, and (c) the standard set augmented by misspecified CFs (designed to challenge the matching model). The findings demonstrate that our full objective is the most robust across these variations, showcasing the adaptability of our method. This experiment underscores a key aspect of our paper: causality is crucial for learning faithful explanations.
>
> 2. Using our novel dataset (section 5.1 and Appendix D), we explored scenarios where the candidate set consists of out-of-distribution texts from topics that substantially differ from those of the primary examples (e.g., abortion vs climate change). Here, our method proved to be the most effective among matching methods, indicating its robustness even in less ideal conditions.
>
> 3. The first row of Table 2 (main results) shows our method's performance when the candidate set is augmented by ground-truth CFs. In this setup, our matching method surpasses even the generative models, illustrating its exceptional effectiveness under optimal conditions.
>
> **Regarding niche tasks with unknown concepts:**
>
> This paper focuses on model interpretability via causal effect estimation (i.e., providing an accurate estimate of how a change of a given variable impacts the prediction of the model). In the causal effect estimation setup, we assume the existence of a causal graph (typically provided by domain experts), outlining the concepts and their interrelations. This assumption might require some effort of concept and causal graph specification, but it is a **requirement** for **accurate causal estimation** (see Shpitser and Pearl 2008 and Feder et.al. 2021 in the references comment). Moreover, as we theoretically show in the main theorem of the paper, a lack of information about the candidate concepts and the causal graph may result in an unfaithful explanation. Hence, we consider the predefined concepts a feature, rather than a limitation of our approach.
>
> This aligns with prevalent practices in causal inference in NLP (see references comment).
> Additionally, pre-defined concepts (properties, attributes, aspects) are extensively used in the NLP model interpretability literature; for example, almost every probing, fairness, and bias detection paper assumes their existence (see references comment). We recognize the significance of this discussion, and we plan to include it in the final version of the paper.
>
> **Regarding open-ended tasks:**
>
> This is a great question. Our framework is not applicable to open-ended tasks (and generation tasks) since the causal effect estimators assume a classifier. We are not aware of any work that aims to address causal model explanations for non-classification tasks, and we consider this an important topic for future research.
>
> **Regarding nested counterfactuals:**
>
> If  “nested counterfactuals” refers to an intervention on a concept that is a parent of a mediator (e.g., given A->B->C, a change in A causes a change in B that changes C), then unfortunately, CEBaB - which is the only causal-driven benchmark for explanation methods - does not support such structures. Notice, however, that this work is the first step for further research, and the ideas proposed by the reviewer indeed provide great directions for future work. We believe that this paper laid the foundation for this kind of research: we show theoretically and empirically that faithful explanations must be causal.
>
> Moreover, we demonstrate that LLMs can be utilized for constructing benchmarks for causal explanation methods that support your ideas: complex causal graphs, many classes, and nested CFs.

---

> ### Author Response · Authors · 2023-11-15
>
> **Update:**
>
> Please see our general comment about updates to the manuscript. Specifically, we added subsections discussing the pre-defined concept requirement and provided examples of CFs generated by LLMs in the medical domain as well as for causal graphs beyond CEBaB. Furthermore, our additional experiments show that our causal matching method is equally effective in a completely unsupervised setup - where the labels are predicted using an LLM instead of human annotations.
>
> **Regarding your question (difference between Random match and Approx):**
>
> The difference between the two methods is that in the approx method, using concept classifiers trained on the training set, we first identify only candidates that share the same value of concepts as the examples and then randomly select a match from them.
> For example, if the intervention is Food (Positive -> Negative), and the remaining three values of the confounders are Ambiance (Negative), Noise (Negative), and Service (Positive) - We first find all candidates with values of Food (Negative), Ambiance (Negative), Noise (Negative), and Service (Positive), and then we randomly select a match from them. In Random Match, we do not perform the identification step, and only randomly select a match from the candidate set (in our example: texts with Food (Negative)).
>
> **References:**
>
> Shpitser Ilya and Pearl Judea. Complete identification methods for the causal hierarchy. Journal of Machine Learning Research, 9, 1941-1979., 2008.
>
> Feder, Amir, et al. Causal inference in natural language processing: Estimation, prediction, interpretation and beyond. Transactions of the Association for Computational Linguistics, 2022, 10: 1138-1158.‏
>
>
> Jesse Vig, Sebastian Gehrmann, Yonatan Belinkov, Sharon Qian, Daniel Nevo, Yaron Singer, and Stuart M. Shieber. Investigating gender bias in language models using causal mediation analysis. In Hugo Larochelle, Marc’Aurelio Ranzato, Raia Hadsell, Maria-Florina Balcan, and Hsuan-Tien Lin (eds.), Advances in Neural Information Processing Systems 33: Annual Conference on Neural Information Processing Systems 2020, NeurIPS 2020, December 6-12, 2020, virtual, 2020. URL https://proceedings.neurips.cc/paper/2020/hash/
> 92650b2e92217715fe312e6fa7b90d82-Abstract.html.
>
> Amir Feder, Nadav Oved, Uri Shalit, and Roi Reichart. CausaLM: Causal model explanation through counterfactual language models. Computational Linguistics, 47(2):333–386, June 2021. doi:10.1162/coli a 00404. URL https://aclanthology.org/2021.cl-2.13.
>
> Divyansh Kaushik, Eduard Hovy, and Zachary C. Lipton. Learning the Difference that Makes a Difference with Counterfactually-Augmented Data. arXiv:1909.12434 [cs, stat], February 2020. URL http://arxiv.org/abs/1909.12434. arXiv: 1909.12434.
>
> Belinkov, Y. (2022). Probing classifiers: Promises, shortcomings, and advances. Computational Linguistics, 48(1), 207-219.‏
>
> Elazar, Yanai, et al. Estimating the Causal Effect of Early ArXiving on Paper Acceptance. arXiv preprint arXiv:2306.13891, 2023.‏
>
> Zhang, Justine; Mullainathan, Sendhil; Danescu-Niculescu-Mizil, Cristian. Quantifying the causal effects of conversational tendencies. Proceedings of the ACM on Human-Computer Interaction, 2020, 4.CSCW2: 1-24.‏
>
> Wood-Doughty, Zach; Sphitser, Ilya; Dredze, Mark. Challenges of using text classifiers for causal inference. In: Proceedings of the Conference on Empirical Methods in Natural Language Processing. Conference on Empirical Methods in Natural Language Processing. NIH Public Access, 2018. p. 4586.‏
>
> Elazar, Yanai, et al. Measuring Causal Effects of Data Statistics on Language Model'sFactual'Predictions. arXiv preprint arXiv:2207.14251, 2022.‏

---

> ### Author Response · Authors · 2023-11-18
>
> **Another answer to the "nested counterfactuals" concern**:
>
> First, our theorem is designed to be flexible and does not rely on a specific type of causal graph, thus, it holds true regardless of the complexity of the causal graph.
>
> Second, consider a diamond structure causal graph consisting of the path A->B->T and the path A->C->T. A nested counterfactual case occurs with A as the treatment since it would influence the mediators B and C and the outcome variable  T. Accordingly, both B and C should not be fixed, and this can be implemented by informing the LLM about the causal structure using its prompt, for example: “Generate a counterfactual by changing the value of the concept A. Notice that A impacts the mediator concepts B and C, which should not be necessarily fixed due to their potential change in response to A”. See the example in the comment below.
>
> This exemplifies that even in the case of nested counterfactuals (when one concept causes a change to another concept) can be modelled with our methods.
>
> The reason we did not explore such scenarios in our paper is that CEBaB does not support them. Yet, we strongly believe that modern strong LLMs like GPT-4 support the generation of such counterfactuals (See the example in the comment below.). Our paper underscores that LLMs can facilitate the creation of new interventional datasets, also ones that are more sophisticated than CEBaB, significantly advancing the research in causal explanations and benchmark construction.
>
> **Example**
> Here is an example of a diamond structure causal graph and a counterfactual generated by GPT-4. The first path is: 'Party Affiliation' -> 'Sympathy toward Trump' ->'Text'. Simultaneously, the second path is 'Party Affiliation' -> 'Sympathy toward Nancy Pelosi' -> ‘Text’.
>
> Notice that this is a simplified example, as there may be many more variables in the causal graph (for example, education could be a confounder as it might impact the affiliation and also the writing style which impacts the text). Nevertheless, the party affiliation should not directly impact the text, but only through mediators (in our example, sympathy toward Trump and Pelosi).
>
> The text is taken from Feder et al. 2022:
>
> > "It was a masterful performance – but behind the sunny smile was the same old Trump: petty, angry, vindictive and deceptive. He refused to shake the hand of House Speaker Nancy Pelosi, a snub she returned in kind by ostentatiously ripping up her copy of the President’s speech at the conclusion of the address, in full view of the cameras."
>
> We prompt GPT-4 to:
> >“Generate a counterfactual by changing the writer's affiliation to the republican party. Notice that the affiliation impacts the sympathy of the writer toward Trump and Pelosi, which should not necessarily be fixed due to their potential change in response to the affiliation.”
>
> The following counterfactual was generated by GPT-4:
> >“It was a decisive and strong performance – behind President Trump's confident demeanor was the same resilient leader: focused, assertive, and forthright. He chose not to shake hands with House Speaker Nancy Pelosi, a decision she responded to with disrespect by theatrically tearing up her copy of the President’s speech at the conclusion of the address, in full view of the cameras.”
>
> This outcome indicates that GPT-4 maintained the value of the mediators ('Sympathy toward Trump' and 'Sympathy toward Pelosi') consistent with the changed treatment concept ('Party Affiliation'). This example demonstrates the capability of LLMs in supporting the generation of nested counterfactuals.

---

> > ### Comment · Reviewer_wvZe · 2023-11-22
> > **Thanks for your comments**
> >
> > On Ablations: this doesn’t address my underlying premise that errors from matching could be reinforced here and will be bottlenecks.
> >
> > On use of novel dataset and OOD results: thanks for the flag, have you by any chance checked what's the shared support between the datasets? If not, that's okay.
> >
> > Table 2: I believe we are in agreement here that the method can outperform in ideal settings, but how realistic are these ideal settings?
> >
> > Appreciate the thoughts on the rest, I'm happy to update the score.

---

> ### Author Response · Authors · 2023-11-22
>
> Thank you for the feedback and discussion and for adjusting our score.
>
> **Regarding your questions:**
>
> **On ablations:**
>
> Our experiments simulate scenarios where the matching set includes misspecified counterfactuals. These counterfactuals result from interventions on both the treated concept and an additional one (e.g., while the intervention was changing the service from positive to negative, the food aspect was also changed from positive to negative). These misspecified CFs. , which bias the estimation (increasing the error) are also highly plausible matches due to their significant semantic overlap with the query (e.g., discussing the same restaurant or menu). This scenario is even more challenging than just injecting noise into the candidate set because the bias we inject is within the most likely match. Notably, although the performance of our matching method declines in this scenario, this degradation is substantially less significant than the degradation in the performance of other ablative models.
>
> **On OOD results:**
>
> We did not check the support between the in-domain and out-domain sets, although we anticipate it to be very low. This is because the sets differ in the topic they discuss (in the stance detection task, given a topic and a text, the model needs to predict whether the text supports, opposes, or is neutral to this topic). For instance, if the query example is about abortion, the candidate set exclusively comprises texts on different topics (although other topics, like feminism, might have a small overlap with abortions). The topics in this setup are Abortion, Atheism, Climate Change, Feminism, and Hilary Clinton.
> Given the low support in these challenging OOD settings, the performance gap between our matching method and the baselines is less pronounced than in the CEBaB dataset. Nonetheless, our method continues to outperform the baseline, showcasing greater robustness.
>
>
> **On Table 2:**
>
> We acknowledge that ideal settings - where the matching set for a given example contains its counterfactual (CF) - are not realistic (yet, the likelihood of finding CF-similar candidates increases with the expansion of the candidate set). These ideal settings serve as an upper bound for our matching method. Nevertheless, we have rigorously tested our method in more challenging and realistic settings, such as with matching sets that include misspecified counterfactuals (“confusing CFs”), OOD, and in all of our experiments the matching sets are relatively limited in size. For instance, in the CEBaB dataset, the whole matching set comprises fewer than a thousand examples, and for a given intervention only about three hundred potentially align with a new treatment value. We believe these realistically reflect practical constraints. Furthermore, when the target domain is known, LLMs can be leveraged to augment the matching set with additional examples. This strategy enables a more comprehensive coverage of the target domain distribution.

---

### Official Review · Reviewer_32rU · 2023-11-02

**Soundness:** 3 good
**Presentation:** 3 good
**Contribution:** 3 good
**Rating:** 6
**Confidence:** 5

**Summary:**

This paper investigates the use of large language models (LLMs) for creating counterfactual examples to explain NLP classification model predictions. Two methods are proposed. The first involves directly using LLMs to generate counterfactual examples, altering only one aspect/concept of the input while maintaining the rest. The second method entails a matching process to discover approximate counterfactuals from a pre-defined candidate set. The study reveals that the matching approach, utilizing a specially trained feature extractor, outperforms strategies using pre-trained LMs as feature extractors and other baselines in the CEBaB benchmark.

------------------------------------
Update after discussion with authors:

The discussion with the authors and the information provided in the updated manuscript made me more convinced of the applicability of the proposed methods, so I raised my score to 6.

**Strengths:**

1. The paper is overall well-written, with a well-defined research question;


2. In addition to direct counterfactual generation via the LLM, the authors introduce an efficient, matching-based approach for identifying approximate counterfactuals from a pre-defined candidate set. Though this method doesn't perform as well as the direct LLM generation, it outperforms past baseline methods and is considerably more efficient than employing the LLM directly for each instance. The exploration of how to efficiently generate counterfactual examples using LLMs is an intriguing aspect of the paper.

**Weaknesses:**

1.

The paper mostly follows the setting of the work of CEBaB, including causal analysis, the approximated counterfactual method, and the evaluation. While some theoretical analysis is provided in Section 3.1, it mainly argues why the approximated counterfactual method which is initially proposed in the CEBaB is better than others. Underlining this, the first concern is that the paper's contribution appears to be limited to the proposal of two LLM-based approximated counterfactual methods that perform better in CEBaB's causal framework. Given the powerful ability of LLM, using it can better generate counterfactual examples (that only change one concept of the input while keep other aspects unchanged) is not very surprising.

The second concern is the limited applicability of the proposed matching method. The use of the matching method under the CEBaB setting requires pre-defined or pre-identified concepts/factors, such as Food (F), Service (S), Ambiance (A), and Noise (N) in restaurant reviews. However, these concepts may not always be available or readily identified in many real-world NLP scenarios. Given that the paper exclusively focuses on using LLMs to generate counterfactual examples in this particular setting, its broader applicability and contribution are questioned.

2.

As indicated by the results in Table 2, the matching method (causal model) demonstrates only a slight improvement over the Approx baseline, especially when K=10. While the matching method is the most novel part of the paper IMO, the fact that its performance isn't significantly superior to the Approx baseline brings its practical importance into question.

In summary, the marginal performance improvement and the limited applicability of the
proposed methods make me tend to reject the paper at this moment.

However, I am open to further discussions and potential rebuttals from the authors that may address these concerns.

**Questions:**

See weaknesses.

**Details Of Ethics Concerns:**

I do not find any particular ethics concerns.

---

> ### Author Response · Authors · 2023-11-15
>
> ‏Thank you for your valuable feedback. We appreciate your openness to further discussions.
>
> **Regarding the contribution of the work (first concern):**
>
> First, we use the term “approximated counterfactual explanation method” as an umbrella term for any explanation in our paper that is based on CFs. The Approx method introduced in CEBaB is just an example of such a method and is a naive adaptation of low-dimensional matching that is abundant in the literature on causal analysis. It only looks for examples with similar concept values, completely ignoring the semantics of the text, which is very different from our approaches.
>
> Second, the theoretical analysis in Section 3.1 explains why any approximated CF explanation method is order-faithful, not only the naive Approx baseline from CEBaB, and underscores the importance of the causal approach to the explanations.
>
> We would like to list the contributions of the paper, which we believe go well beyond the reviewer’s statement:
>
> 1. While the idea of causal explanations being integral to model faithfulness has been discussed in prior works, including CEBaB, our paper is the first to provide a theoretical framework that unifies the concepts of faithfulness and causal explanation. Specifically, we prove that CFs (which are a fundamental concept in causal inference) guarantee faithfulness, unlike other model explanation techniques, such as probing, concept removal, adversarial approaches, etc.
>
> 2. In line with recent advancements in the capabilities of LLMs, we are the first to show that LLM-generated CFs provide the SOTA NLP model explanation.
>
> 3. Moreover, we demonstrate empirically that utilizing multiple CFs enhances the quality of the causal estimator across various explanation methodologies. This empirical validation is the first of its kind and is both important and useful.
>
> 4. This is the first paper that thoroughly explores matching techniques for causal model explanation in NLP. Specifically, we enhance the matching paradigm originating in  the causal literature such that it considers the rich representation of the text, and not only the observed concepts. This key contribution lays the groundwork for future research in causal representation learning from text.
>
> 5. Finally, recognizing the gap in benchmarks for causal model explanations, we construct a new interventional benchmark. This demonstration paves the way for future research to explore causal model explanations for a broader range of NLP problems and more complex setups.
>
> In essence, our main contribution lies in theoretically and empirically establishing that faithful explanation methods must be causal and propose CFs as the central mechanism to achieve this. We believe that this insight is pivotal for the field of model interpretability.
>
>
> **Regarding the pre-defined concepts:**
>
> This paper focuses on model interpretability via causal effect estimation (i.e., providing an accurate estimation of how a change of a given variable impacts the prediction of the model). This differs from other model interpretability problems, such as finding concepts that  may correlate with the prediction.
>
> Accordingly, in the causal effect estimation setup, we assume the existence of a causal graph (typically provided by domain experts), outlining the concepts and their interrelations. This assumption might require some effort of concept and causal graph specification, but it is a **requirement** for **accurate causal estimation** (see Shpitser and Pearl 2008 and Feder et.al. 2021 in the references comment). Moreover, as we theoretically show in the main theorem of the paper, a lack of information about the candidate concepts and the causal graph may result in an unfaithful explanation. Hence, we consider the predefined concepts a feature, rather than a limitation of our approach.
>
> This aligns with prevalent practices in causal inference in NLP (Feder et al., 2022, see references comment).  Additionally, pre-defined concepts (properties, attributes, aspects) are extensively used in the NLP model interpretability literature; for example, almost every probing, fairness, and bias detection paper assumes their existence (see references comment). We recognize the importance of this discussion, and we plan to include it in the final version of the paper.

---

> > ### Author Response · Authors · 2023-11-15
> >
> > **Regarding concept annotations:**
> >
> > Although we used concept annotations in our experiments, we believe they are not required and can be easily inferred by an LLM.
> > Moreover, we would like to point to experiments from our ablation study (Appendix C, referred to on pages 5 and 7), which offers valuable insights into the applicability of our matching approach. In Table 4, rows 3 and 8 present experiments where components that rely on pre-identified concepts are omitted.
> > Row 3 presents an experiment where misspecified CFs are not filtered. This filtering relies on concept classifiers, which are trained with concept annotations.
> >
> > Row 8 presents an experiment without the matches (X_M) and misspecified matches (X_~M) sets, which are part of the optimization objective and are constructed using concept annotations. We found that these omissions had minimal performance impact. However, this changes markedly when the quality of the candidate set changes. When the candidate set is augmented with misspecified CFs (“confusing examples”), models trained without these components show a noticeable decline in performance compared to the full model. This experiment underscores a key observation of our paper: a definition of the causal mechanism is crucial for learning faithful explanations, and without pre-defined concepts, explanation methods might default to mere correlational insights, which, while potentially effective, lack robustness. In challenging scenarios, such methods are prone to failure, as demonstrated in our main theorem: A non-causal method can be faithful for a given problem, but small changes to the causal mechanism may make it unfaithful.
> >
> >
> > **Regarding the results of the matching method:**
> >
> > We argue that while the matching method is the main methodological contribution of this paper, the contributions of the paper go well beyond this contribution (please see the above comment about contributions). We would like to highlight the following aspects of the comparison between our matching method and the Approx baseline:
> >
> > 1. In Table 2: Our matching method consistently outperforms the Approx baseline for every metric and every model. Moreover, each number in the table is an average of 12 interventions, and while this is not shown in the table, we confirm that our method is superior in each of them.
> >
> > 2. Our matching method is also substantially faster (7-28 times faster, see Appendix B) and does not require concept classifiers like Approx.
> >
> > 3. Our matching method surpasses the generative models in scenarios where the candidate set includes ground-truth CFs (as shown in the first row of Table 2).
> >
> > 4. The comparative analysis presented in Figures 2 and 3 illustrates the superiority of our matching method over the Approx baseline. Our method accurately ranks candidates (the lower the rank, the bigger the estimation error is) - unlike other matching baselines.
> >
> > 5. Regarding the comment about K matches: This technique improves all the methods we examined (generative and matching), which is also an important finding of our paper. Moreover, the Approx baseline samples a match from a subset of candidates agreeing on all the evaluated example's concept values. Therefore, it might not be applicable if this subset is empty or small, which could happen when there is a large number of concepts or when some concept values have a particularly low probability (although in CEBaB this did not happen).
> >
> > **References:**
> >
> > Shpitser Ilya and Pearl Judea. Complete identification methods for the causal hierarchy. Journal of Machine Learning Research, 9, 1941-1979., 2008.
> >
> > Feder, Amir, et al. Causal inference in natural language processing: Estimation, prediction, interpretation and beyond. Transactions of the Association for Computational Linguistics, 2022, 10: 1138-1158.
> >
> > Jesse Vig, et al. Investigating gender bias in language models using causal mediation analysis. In NeurIPS 2020, December 6-12, 2020.
> >
> > Amir Feder, Nadav Oved, Uri Shalit, and Roi Reichart. CausaLM: Causal model explanation through counterfactual language models. Computational Linguistics, 47(2):333–386, June 2021.
> >
> > Belinkov, Y. (2022). Probing classifiers: Promises, shortcomings, and advances. Computational Linguistics, 48(1), 207-219.
> >
> > Elazar, Yanai, et al. Estimating the Causal Effect of Early ArXiving on Paper Acceptance. arXiv preprint arXiv:2306.13891, 2023.
> >
> > Zhang, Justine; Mullainathan, Sendhil; Danescu-Niculescu-Mizil, Cristian. Quantifying the causal effects of conversational tendencies. Proceedings of the ACM on Human-Computer Interaction, 2020, 4.CSCW2: 1-24.
> >
> > Wood-Doughty, Zach; Sphitser, Ilya; Dredze, Mark. Challenges of using text classifiers for causal inference. In: Proceedings of the Conference on Empirical Methods in Natural Language Processing. 2018. p. 4586.
> >
> > Elazar, Yanai, et al. Measuring Causal Effects of Data Statistics on Language Model'sFactual'Predictions. arXiv preprint arXiv:2207.14251, 2022.

---

> > > ### Comment · Reviewer_32rU · 2023-11-17
> > > **Response to Rebuttal**
> > >
> > > Thanks for the authors’ responses. I now understand the paper’s contribution better.
> > >
> > > However, after reading other reviewer’s reviews, I found that we actually have some shared concerns of this paper. Reviewer 9xHf pointed out that “the concepts are pre-defined, which can be a limiting factor” and Reviewer wvZe said “such attributes may be hard to define beforehand”. While authors provided responses to those questions, I still have multiple concerns:
> > >
> > > 1.	The causal model the model based on is a simplified one where each concept is independent of the other. I suspect that it is not the general case in practice. Would the theoretical deduction still hold when it is not an independent case? Or how easy/difficult does current theory discussion generate to the more complex causal graph? This point is related to the “nested counterfactuals” mentioned by Reviewer wvZe.
> > > 2.	Regarding the “pre-defined concepts”. I understand in a causal effect estimation setup, the pre-defined concept is a requirement. As mentioned by the authors, such concepts are usually defined by domain experts (humans). A question here is that, the model does not necessarily rely on the concepts humans believe as important to make predictions. Some model biases can be imagined by humans, but some are not. In such a case, we can only verify whether there are model biases towards some particular listed concepts, but it is still far away from understating the whole model behaviors including its hidden biases. Authors mentioned “Although we used concept annotations in our experiments, we believe they are not required and can be easily inferred by an LLM”, I am wondering here the authors mean we can use the LLM to list some concepts like domain experts, or we can use the LLM to explore some hidden concepts based on the training data. If it is the second case, I am happy to see some examples.
> > >
> > > In summary, while I acknowledge the contributions paper has made within the current causal framework, including the theoretical deduction, proposed LLM-based methods and the new benchmark, I still have reservations about how well the simplified causal model aligns with practical explanation cases. I will wait for the authors’ further responses before reconsidering my rating.

---

> > > > ### Author Response · Authors · 2023-11-17
> > > >
> > > > Thank you for your engagement with our work and for raising these questions. Some of your concerns present valuable directions for future research.
> > > >
> > > > **First concern:**
> > > >
> > > > Our theorem is designed to be flexible and does not rely on a specific type of causal graph. Our proof is based on a minor modification to the graph, adding a confounder, which can be applied to graphs of any complexity. The key observation is that while non-causal methods may struggle with this change, causal methods remain faithful. Thus, our theorem holds true regardless of the complexity of the causal graph.
> > > >
> > > > Regarding nested counterfactuals: Our methods and the theorem are still relevant in this case. To realise this, consider a diamond structure causal graph, consisting of the path A->B->T and the path A->C->T. A nested counterfactual case occurs with A as the treatment since it would influence the mediators B and C and the outcome variable  T. Accordingly, both B and C should not be fixed, and this can be implemented by informing the LLM about the causal structure using its prompt, for example: “Generate a counterfactual by changing the value of the concept A. Notice that A impacts the mediator concepts B and C, which should not be necessarily fixed due to their potential change in response to A”. See the example in the comment below.
> > > > This exemplifies that even in the case of nested counterfactuals (when one concept causes a change to another concept) can be modeled with our methods.
> > > >
> > > > We did not explore such scenarios in our paper because CEBaB does not support them. Yet, we strongly believe that strong modern LLMs like GPT-4 support the generation of such counterfactuals (See the example in the comment below). Our paper underscores that LLMs can facilitate the creation of new interventional datasets, also ones that are more sophisticated than CEBaB, significantly advancing the research in causal explanations and benchmark construction.
> > > >
> > > > We thank the reviewer for this excellent comment and admit that we should have addressed it in the original version of the paper. We will add a clarification to the Appendix during the discussion period and it will be part of the final version of the paper. We will also mention this direction as an important direction of future research in the main text of the paper.
> > > >
> > > >
> > > > **Second concern:**
> > > >
> > > > It is noteworthy that in some fields, causal explanations are crucial, making the effort to construct a causal graph a worthwhile endeavour. Consider an NLP model that recommends medical treatments based on symptoms described in the medical record. In that case, the clinician can not rely on correlational interpretations of the model recommendations.
> > > >
> > > > There are a few preliminary works that evaluate the capabilities of LLMs to construct a causal graph, for example, Kiciman et al. 2023, Arsenyan et al. 2023 and Tu et al. 2023 (see references  below). While these papers do not rely on training data but on world knowledge acquired by LLMs, constructing a causal graph from training data (observational data) using an LLM is a promising direction. In the very recent paper of Ludan et al. 2023, the authors augment the prompt of an LLM with examples from the training data and ask which concepts might impact the label. Notice, however, that this is not a causal approach (since no causal structure is discovered, only concepts) and it also does not attempt to explain model predictions (the authors develop an interpretable model based on the discovered concepts). Nevertheless, this demonstrates the promising capabilities of LLMs for such tasks. Following future research and technological advances, it is reasonable to assume that the reliance on human experts for concept and causal graph discovery will reduce, and this process will gradually become more and more automatic. Hopefully, our paper will contribute to the effort of unifying causality and model interpretability and facilitate important research in the directions suggested by the reviewer.
> > > >
> > > > **References:**
> > > >
> > > > Kıcıman, E., Ness, R., Sharma, A., & Tan, C. (2023). Causal reasoning and large language models: Opening a new frontier for causality. arXiv preprint arXiv:2305.00050.‏
> > > >
> > > > Arsenyan, V., & Shahnazaryan, D. (2023). Large Language Models for Biomedical Causal Graph Construction. arXiv preprint arXiv:2301.12473.‏
> > > >
> > > > Tu, R., Ma, C., & Zhang, C. (2023). Causal-discovery performance of chatgpt in the context of neuropathic pain diagnosis. arXiv preprint arXiv:2301.13819.‏
> > > >
> > > > Ludan, J. M., Lyu, Q., Yang, Y., Dugan, L., Yatskar, M., & Callison-Burch, C. (2023). Interpretable-by-Design Text Classification with Iteratively Generated Concept Bottleneck. arXiv preprint arXiv:2310.19660.‏

---

> ### Author Response · Authors · 2023-11-17
>
> Here is an example of a diamond structure causal graph and a counterfactual generated by GPT-4. The first path is: 'Party Affiliation' -> 'Sympathy toward Trump' ->'Text'. Simultaneously, the second path is 'Party Affiliation' -> 'Sympathy toward Nancy Pelosi' -> ‘Text’.
>
> Notice that this is a simplified example, as there may be many more variables in the causal graph (for example, education could be a confounder as it might impact the affiliation and also the writing style, which impacts the text). Nevertheless, the party affiliation should not directly impact the text but only through mediators (in our example, sympathy toward Trump and Pelosi).
>
> The text is taken from Feder et al. 2022:
>
> > "It was a masterful performance – but behind the sunny smile was the same old Trump: petty, angry, vindictive and deceptive. He refused to shake the hand of House Speaker Nancy Pelosi, a snub she returned in kind by ostentatiously ripping up her copy of the President’s speech at the conclusion of the address, in full view of the cameras."
>
> We prompt GPT-4 to:
> > “Generate a counterfactual by changing the writer's affiliation to the republican party. Notice that the affiliation impacts the sympathy of the writer toward Trump and Pelosi, which should not necessarily be fixed due to their potential change in response to the affiliation.”
>
> The following counterfactual was generated by GPT-4:
> > “It was a decisive and strong performance – behind President Trump's confident demeanor was the same resilient leader: focused, assertive, and forthright. He chose not to shake hands with House Speaker Nancy Pelosi, a decision she responded to with disrespect by theatrically tearing up her copy of the President’s speech at the conclusion of the address, in full view of the cameras.”
>
> This outcome indicates that GPT-4 maintained the value of the mediators ('Sympathy toward Trump' and 'Sympathy toward Pelosi') consistent with the changed treatment concept ('Party Affiliation'). This example demonstrates the capability of LLMs to support the generation of nested counterfactuals.

---

> > ### Comment · Reviewer_32rU · 2023-11-19
> >
> > Thanks for the authors' further responses.
> >
> > At this moment, while I now understand the paper's contributions based on the CEBaB framework, I remain concerned about its limited applicability. As the authors suggested in their responses, there's potential to enhance the framework and methods using powerful modern LLMs like GPT-4. I encourage the authors to delve deeper into this, in the future version of this paper (in ICLR2024 or elsewhere).
> >
> > Also, is there a mistake somewhere in the example provided? I checked the original text and the generated counterfactual verbatim and found them to be identical.  When I used GPT-4 with the same text and prompt, it generated a counterfactual that differed from the original text. Additionally, I would recommend the authors to use examples from fields other than politics.
> >
> > I will raise my score to 5. The increase reflects my recognition of the paper's contributions, but concerns over its limited applicability prevent me from rating it higher.

---

> ### Author Response · Authors · 2023-11-19
>
> Thank you for your engagement in this discussion.
>
> You are correct about the example - there was an error. We inadvertently pasted the counterfactual instead of the original example. We have now edited the comment and fixed this.
>
> In addition, we acknowledge your feedback concerning the use of political examples. We plan to add an example from the medical domain to our comments and to a section in the appendix that discusses complex causal graphs with mediators.
>
> **Update:**
>
> Please see our general comment about updates to the manuscript. Specifically, we added subsections discussing the pre-defined concept requirement and provided examples of CFs generated by LLMs in the medical domain as well as for causal graphs beyond CEBaB. Furthermore, our additional experiments show that our causal matching method is equally effective in a completely unsupervised setup - where the labels are predicted using an LLM instead of human annotations.

---

> > ### Comment · Reviewer_32rU · 2023-11-22
> >
> > The information provided in the updated manuscript makes me more convinced of the applicability of the proposed methods, so I raised my score to 6.

---

### Official Review · Reviewer_9xHf · 2023-11-07

**Soundness:** 3 good
**Presentation:** 4 excellent
**Contribution:** 3 good
**Rating:** 6
**Confidence:** 4

**Summary:**

The paper proposes two methods for approximating Counterfactuals in a model-agnostic way. The first one is to utilize an LLM to change attributes during inference time. The second method is to find Counterfactuals through efficient matching. In order to allow efficient matching, the paper developed a novel language representation learning method specifically for encoding counterfactuals. Such representation is learned through contrastive loss that maximizes the similarity of approximate counterfactuals and minimizes similarities of misspecified Counterfactuals. Both methods achieve better performances than prior works. The paper also released a dataset for evaluating NLP explanation techniques.

**Strengths:**

1. The paper proposed an efficient and novel matching technique for finding Counterfactuals and provided strong theoretical and practical evidence that Counterfactuals are good explanations.
2. Counterfactuals generated using this method are more order-faithful and comprehensive than prior work.
3. Detailed ablation study to demonstrate the effect of each component in the method.

**Weaknesses:**

1. The results are only on one dataset CEBaB. The experimental section would be more convincing if more experiments were done on a wider range of datasets.
2. The concepts are pre-defined, which can be a limiting factor to the comprehensiveness of the Counterfactuals generated.

**Questions:**

1. If the matching candidate set doesn't exist, do you generate them given the concepts?

---

> ### Author Response · Authors · 2023-11-15
>
> Thank you for the insightful feedback and constructive comments.
>
> **Regarding using only the CEBaB dataset:**
>
> Our choice of CEBaB was driven by its unique status as the **only** non-synthetic interventional dataset tailored for benchmarking concept-level explanation methods in NLP. While automatic techniques for evaluating explanations exist and can be applied to non-interventional datasets, they predominantly assess correlation rather than causation (as discussed in our Introduction) and, therefore, are unsuitable for this paper.
>
> Nevertheless, recognizing this limitation, we have not limited our investigation to CEBaB only. As detailed in subsection 5.1 and Appendix D, we introduced a new benchmark focusing on Stance Detection. This extension serves to validate our findings in diverse contexts, demonstrating the replicability of our approach.
>
> Additionally, our work underscores a novel contribution in utilizing LLMs for generating Counterfactuals (CFs). This, as we demonstrated, can facilitate the creation of new interventional datasets, significantly advancing the research in causal explanations and benchmark construction. We plan to highlight these aspects more prominently in the revised manuscript.
>
> **Regarding pre-defined concepts:**
>
> This paper focuses on model interpretability via causal effect estimation (i.e., providing an accurate estimation of how a change of a given variable impacts the prediction of the model). This differs from other model interpretability problems, such as finding concepts that  may correlate with the prediction.
>
> Accordingly, in the causal effect estimation setup, we assume the existence of a causal graph (typically provided by domain experts), outlining the concepts and their interrelations. This assumption might require some effort of concept and causal graph specification, but it is a **requirement** for **accurate causal estimation** (see Shpitser and Pearl 2008 and Feder et.al. 2021 in the references comment). Moreover, as we theoretically show in the main theorem of the paper, a lack of information about the candidate concepts and the causal graph may result in an unfaithful explanation. Hence, we consider the predefined concepts a feature, rather than a limitation of our approach.
>
> This aligns with prevalent practices in causal inference in NLP (Feder et al., 2022, see references comment).  Additionally, pre-defined concepts (properties, attributes, aspects) are extensively used in the NLP model interpretability literature; for example, almost every probing, fairness, and bias detection paper assumes their existence (see references comment). We recognize the importance of this discussion, and we plan to include it in the final version of the paper.
>
> **Regarding your question:**
>
> As mentioned by the reviewer, we could generate the candidate set using LLMs, although this is not what we did in practice. We constructed the candidate sets as follows:
> For the CEBaB setup, the candidate set is randomly sampled from the training set. Notice that this candidate set is used only at inference time.
> For the novel Stance Detection setup, we rely on texts from the SemEval16 dataset but generate the entire training, dev, test, and candidate sets.
>
> The only assumptions we make in this paper are about the pre-defined concepts and the causal graph, and the existence of training and dev sets. The CFs we use for training the models are generated by an LLM, making our matching approach more general compared to the alternative of using human-written CFs.
>
> **Update:**
>
> Please see our general comment about updates to the manuscript. Specifically, we added subsections discussing the pre-defined concept requirement and provided examples of CFs generated by LLMs in the medical domain as well as for causal graphs beyond CEBaB.
> Furthermore, our additional experiments show that our causal matching method is equally effective in a completely unsupervised setup - where the labels are predicted using an LLM instead of human annotations.

---

> > ### Author Response · Authors · 2023-11-15
> >
> > **References:**
> >
> > Shpitser Ilya and Pearl Judea. Complete identification methods for the causal hierarchy. Journal of Machine Learning Research, 9, 1941-1979., 2008.
> >
> > Feder, Amir, et al. Causal inference in natural language processing: Estimation, prediction, interpretation and beyond. Transactions of the Association for Computational Linguistics, 2022, 10: 1138-1158.‏
> >
> >
> > Jesse Vig, Sebastian Gehrmann, Yonatan Belinkov, Sharon Qian, Daniel Nevo, Yaron Singer, and Stuart M. Shieber. Investigating gender bias in language models using causal mediation analysis. In Hugo Larochelle, Marc’Aurelio Ranzato, Raia Hadsell, Maria-Florina Balcan, and Hsuan-Tien Lin (eds.), Advances in Neural Information Processing Systems 33: Annual Conference on Neural Information Processing Systems 2020, NeurIPS 2020, December 6-12, 2020, virtual, 2020. URL https://proceedings.neurips.cc/paper/2020/hash/
> > 92650b2e92217715fe312e6fa7b90d82-Abstract.html.
> >
> > Amir Feder, Nadav Oved, Uri Shalit, and Roi Reichart. CausaLM: Causal model explanation through counterfactual language models. Computational Linguistics, 47(2):333–386, June 2021. doi:10.1162/coli a 00404. URL https://aclanthology.org/2021.cl-2.13.
> >
> > Divyansh Kaushik, Eduard Hovy, and Zachary C. Lipton. Learning the Difference that Makes a Difference with Counterfactually-Augmented Data. arXiv:1909.12434 [cs, stat], February 2020. URL http://arxiv.org/abs/1909.12434. arXiv: 1909.12434.
> >
> > Belinkov, Y. (2022). Probing classifiers: Promises, shortcomings, and advances. Computational Linguistics, 48(1), 207-219.‏
> >
> > Elazar, Yanai, et al. Estimating the Causal Effect of Early ArXiving on Paper Acceptance. arXiv preprint arXiv:2306.13891, 2023.‏
> >
> > Zhang, Justine; Mullainathan, Sendhil; Danescu-Niculescu-Mizil, Cristian. Quantifying the causal effects of conversational tendencies. Proceedings of the ACM on Human-Computer Interaction, 2020, 4.CSCW2: 1-24.‏
> >
> > Wood-Doughty, Zach; Sphitser, Ilya; Dredze, Mark. Challenges of using text classifiers for causal inference. In: Proceedings of the Conference on Empirical Methods in Natural Language Processing. Conference on Empirical Methods in Natural Language Processing. NIH Public Access, 2018. p. 4586.‏
> >
> > Elazar, Yanai, et al. Measuring Causal Effects of Data Statistics on Language Model'sFactual'Predictions. arXiv preprint arXiv:2207.14251, 2022.‏

---

> > > ### Comment · Reviewer_9xHf · 2023-11-20
> > >
> > > Thank you for the detailed response from the authors. I saw the authors making great efforts to improve the model's general applicability in the response. I understand that pre-defined concepts are considered features rather than limitations for causal effect analysis. It is also helpful to know that a new stance detection dataset is proposed. I recognize the paper's contribution on improving CFs as model explanations in terms of performance and efficiency.
> > >
> > > However, I still share some of the concerns of other reviewers, e.g. reviewer 32rU. While now I think it is fine to pre-define concepts, such concepts are artificial and doesn't reflect how models actually make such decisions. For example, there are four concepts available in the CEBaB benchmark. But there could be a lot more features and even spurious features that may be more relevant to the input example being analyzed.
> > >
> > > With that said, I still think this is a really interesting research question to be discussed further for the community and the authors did make contributions within the causal analysis framework. I will maintain my score and confidence for this.

---

### Author Response · Authors · 2023-11-20

We thank the reviewers for their feedback.

We have carefully updated our manuscript to comprehensively address the concerns raised by the reviewers. Specifically, we added four subsections to the appendix referred from the main body (from sections 3, 4, 6). In addition, we conducted additional ablation experiments.

**Subsection F.1:**
This subsection details and discusses the setup requirements of our methods, specifically focusing on the causal graph and pre-defined concept specification, as suggested by reviewers 9xHf, 32rU, and wvZe. It includes a discussion on pre-identified concept annotation, supported by additional ablation experiments (see below).

**Subsection F.2:**
Addresses the handling of additional and more complex causal graphs beyond CEBaB (raised by reviewers 9xHf, 32rU, wvZe). Alongside this discussion, we provide a medical domain example demonstrating the capacity of strong LLMs like GPT-4 to generate counterfactuals in scenarios with causal graphs including both confounders and mediators. This section specifically elucidates on mediation where the intervention on one concept also impacts another concept ("nested CFs" as highlighted by wvZe) and delves into the estimation of both direct and total causal effects.

**Subsection G.1 and G.2:**
As recommended by reviewer GEWc, we have introduced a detailed description of the causal estimation pipeline (G.1) and the evaluation pipeline of explanation methods (G.2). These are referenced from Sections 3 and 4, offering an enriched understanding of the methodologies employed.

**New experiments:**
In addition, We have expanded our ablation study to investigate the applicability of our method without pre-identified concept annotations in the training dataset (i.e., completely unsupervised setup). Utilizing ChatGPT (GPT-3.5), we predicted concept values in a zero-shot manner. The findings, as detailed in Section C (Ablation Study), reveal that the performance remains consistent and on par with models trained on human-annotated datasets. Surprisingly, it performs better in some metrics (although insignificant), likely because the LLM predicts annotations that are sometimes missed due to disagreements between annotators.
We hope our new findings satisfy the reviewers regarding the applicability of our method.

---

### Meta-Review · Area_Chair_YsUT · 2023-12-07

**Metareview:**

This paper proposes two approaches to utilize LLMs for counterfactual explanation of NLP models. The first approach directly prompts LLMs to change, in input text, a specific concept while keeping other concepts unchanged.  The second approach is an efficient approximation of the first approach and it requires a candidate set.  The reviewers consider the work a worthy contribution to NLP, with solid theoretical and empirical results. There were some concerns at the start about the requirement of a candidate set and the experiments being run on only one benchmark. After the discussions, the reviewers’ views converged to weak accept.

**Justification For Why Not Higher Score:**

No reviewers are excited about this work.

**Justification For Why Not Lower Score:**

The work seems reasonable.

---

### Decision · Program_Chairs · 2024-01-16

Accept (poster)